# Uncertainty Quantification over Graph with Conformalized Graph Neural Networks

**Kexin Huang**[1]    **Ying Jin**[2]    **Emmanuel Candès**[2,3]    **Jure Leskovec**[1]

[1] Department of Computer Science, Stanford University
[2] Department of Statistics, Stanford University
[3] Department of Mathematics, Stanford University
kexinh@cs.stanford.edu, ying531@stanford.edu,
candes@stanford.edu, jure@cs.stanford.edu

## Abstract

Graph Neural Networks (GNNs) are powerful machine learning prediction models on graph-structured data. However, GNNs lack rigorous uncertainty estimates, limiting their reliable deployment in settings where the cost of errors is significant. We propose conformalized GNN (CF-GNN), extending conformal prediction (CP) to graph-based models for guaranteed uncertainty estimates. Given an entity in the graph, CF-GNN produces a prediction set/interval that provably contains the true label with pre-defined coverage probability (e.g. 90%). We establish a permutation invariance condition that enables the validity of CP on graph data and provide an exact characterization of the test-time coverage. Besides valid coverage, it is crucial to reduce the prediction set size/interval length for practical use. We observe a key connection between non-conformity scores and network structures, which motivates us to develop a topology-aware output correction model that learns to update the prediction and produces more efficient prediction sets/intervals. Extensive experiments show that CF-GNN achieves any pre-defined target marginal coverage while significantly reducing the prediction set/interval size by up to 74% over the baselines. It also empirically achieves satisfactory conditional coverage over various raw and network features.

## 1 Introduction

Graph Neural Networks (GNNs) have shown great potential in learning representations for graph-structured data, which has led to their widespread adoption in weather forecasting [29], drug discovery [31], and recommender systems [46], etc. As GNNs are increasingly deployed in high-stakes settings, it is important to understand the uncertainty in the predictions they produce. One prominent approach to uncertainty quantification is to construct a prediction set/interval that informs a plausible range of values the true outcome may take. A large number of methods have been proposed to achieve this goal [17, 49, 28, 44]. However, these methods often lack theoretical and empirical guarantees regarding their validity, i.e. the probability that the prediction set/interval covers the outcome [2]. This lack of rigor hinders their reliable deployment in situations where errors can be consequential.

Conformal prediction [43] (CP) is a framework for producing statistically guaranteed uncertainty estimates. Given a user-specified miscoverage level $\alpha \in (0, 1)$, it leverages a set of "calibration" data to output prediction sets/intervals for new test points that provably include the true outcome with probability at least $1 - \alpha$. Put another way, the conformal prediction sets provably only miss the test outcomes at most $\alpha$ fraction of the time. With its simple formulation, clear guarantee and

distribution-free nature, it has been successfully applied to various problems in computer vision [2, 4], causal inference [30, 22, 47], time series forecasting [11, 48], and drug discovery [21].

Despite its success in numerous domains, conformal prediction has remained largely unexplored in the context of graph-structured data. One primary challenge is that it is unclear if the only, yet crucial, assumption for CP—exchangeability between the test and calibration samples—holds for graph data. When applying conformal prediction, exchangeability is usually ensured by independence among the trained model, the calibration data, and test samples (see Appendix B for more discussion). However, in the transductive setting, GNN training employs all nodes within the same graph–including test points–for message passing, creating intricate dependencies among them. Thus, to deploy conformal prediction for graph data, the first challenge is to identify situations where valid conformal prediction is possible given a fitted GNN model that already involves test information.

Efficiency is another crucial aspect of conformal prediction for practical use: a prediction set/interval with an enormous set size/interval length might not be practically desirable even though it achieves valid coverage. Therefore, the second major challenge is to develop a graph-specific approach to reduce the size of the prediction set or the length of the prediction interval (dubbed as *inefficiency* hereafter for brevity) while retaining the attractive coverage property of conformal prediction.

**Present work.** We propose conformalized GNN (CF-GNN),[1] extending conformal prediction to GNN for rigorous uncertainty quantification over graphs. We begin by establishing the validity of conformal prediction for graphs. We show that in the transductive setting, regardless of the dependence among calibration, test, and training nodes, standard conformal prediction [43] is valid as long as the score function (whose definition will be made clear in Section 2) is invariant to the ordering of calibration and test samples. This condition is easily satisfied by popular GNN models. Furthermore, we provide an exact characterization of the empirical test-time coverage.

Subsequently, we present a new approach that learns to optimize the inefficiencies of conformal prediction. We conduct an empirical analysis which reveals that inefficiencies are highly correlated along the network edges. Based on this observation, we add a topology-aware correction model that updates the node predictions based on their neighbors. This model is trained by minimizing a differentiable efficiency loss that simulates the CP set sizes/interval lengths. In this way, unlike the raw prediction that is often optimized for prediction accuracy, the corrected GNN prediction is optimized to yield smaller/shorter conformal prediction sets/intervals. Crucially, our approach aligns with the developed theory of graph exchangeability, ensuring valid coverage guarantees while simultaneously enhancing efficiency.

We conduct extensive experiments across 15 diverse datasets for both node classification and regression with 8 uncertainty quantification (UQ) baselines, covering a wide range of application domains. While all previous UQ methods fail to reach pre-defined target coverage, CF-GNN achieves the pre-defined empirical marginal coverage. It also significantly reduces the prediction set sizes/interval lengths by up to 74% compared with a direct application of conformal prediction to GNN. Such improvement in efficiency does not appear to sacrifice adaptivity: we show that CF-GNN achieves strong empirical conditional coverage over various network features.

## 2 Background and Problem Formulation

Let $G = (\mathcal{V}, \mathcal{E}, \mathbf{X})$ be a graph, where $\mathcal{V}$ is a set of nodes, $\mathcal{E}$ is a set of edges, and $\mathbf{X} = \{\mathbf{x}_v\}_{v \in \mathcal{V}}$ is the attributes, where $\mathbf{x}_v \in \mathbb{R}^d$ is a $d$-dimensional feature vector for node $v \in \mathcal{V}$. The label of node $v$ is $y_v \in \mathcal{Y}$. For classification, $\mathcal{Y}$ is the discrete set of possible label classes. For regression, $\mathcal{Y} = \mathbb{R}$.

**Transductive setting.** We focus on transductive node classification/regression problems with random data split. In this setting, the graph $G$ is fixed. At the beginning, we have access to $\{(\mathbf{x}_v, y_v)\}_{v \in \mathcal{D}}$ as the "training" data, as well as test data $\mathcal{D}_{\text{test}}$ with unknown labels $\{y_v\}_{v \in \mathcal{D}_{\text{test}}}$. Here $\mathcal{D}$ and $\mathcal{D}_{\text{test}}$ are disjoint subsets of $\mathcal{V}$. We work on the prevalent random split setting where nodes in $\mathcal{D}$ and $\mathcal{D}_{\text{test}}$ are randomly allocated from the entire graph, and the test sample size is $m = |\mathcal{D}_{\text{test}}|$. The training node set $\mathcal{D}$ is then randomly split into $\mathcal{D}_{\text{train}}/\mathcal{D}_{\text{valid}}/\mathcal{D}_{\text{calib}}$ of fixed sizes, the training/validation/calibration set, correspondingly. A perhaps nonstandard point here is that we withhold a subset $\mathcal{D}_{\text{calib}}$ as "calibration" data in order to apply conformal prediction later on. During the training step, the data

---

[1]See Figure 1 for an overview. The code is available at https://github.com/snap-stanford/conformalized-gnn.

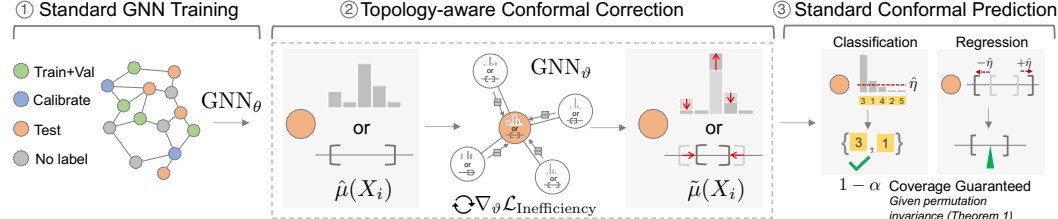

**Figure 1:** Conformal prediction for graph-structured data. (1) GNN training. We first use standard GNN training to obtain a base GNN model (GNN$_\theta$) that produces prediction scores $\widehat{\mu}(X_i)$ for node $i$. It is fixed once trained. (2) Conformal correction. Since the training step is not aware of the conformal calibration step, the size/length of prediction sets/intervals (i.e. efficiency) are not optimized. We propose a novel correction step that learns to correct the prediction to achieve desirable properties such as efficiency. We use a topology-aware correction model GNN$_\vartheta$ that takes $\widehat{\mu}(X_i)$ as the input node feature and aggregates information from its local subgraph to produce an updated prediction $\tilde{\mu}(X_i)$. $\vartheta$ is trained by simulating the conformal prediction step and optimizing a differentiable inefficiency loss. (3) Conformal prediction. We prove that in a transductive random split setting, graph exchangeability holds (Section 3) given permutation invariance. Thus, standard CP can be used to produce a prediction set/interval based on $\tilde{\mu}$ that includes true label with pre-specified coverage rate 1-$\alpha$.

$\{(\mathbf{x}_v, y_v)\}_{v \in \mathcal{D}_{\text{train}} \cup \mathcal{D}_{\text{valid}}}$, the attribute information in $\{\mathbf{x}_v\}_{v \in \mathcal{D}_{\text{calib}} \cup \mathcal{D}_{\text{test}}}$ and the entire graph structure $(\mathcal{V}, \mathcal{E})$ are available to the GNN to compute training nodes representations, while $\{y_v\}_{v \in \mathcal{D}_{\text{calib}} \cup \mathcal{D}_{\text{test}}}$ are not seen.

**Graph Neural Networks (GNNs).** GNNs learn compact representations that capture network structure and node features. A GNN generates outputs through a series of propagation layers [12], where propagation at layer $l$ consists of the following three steps: (1) Neural message passing. GNN computes a message $\mathbf{m}_{uv}^{(l)} = \text{Msg}(\mathbf{h}_u^{(l-1)}, \mathbf{h}_v^{(l-1)})$ for every linked nodes $u, v$ based on their embeddings from the previous layer $\mathbf{h}_u^{(l-1)}$ and $\mathbf{h}_v^{(l-1)}$. (2) Neighborhood aggregation. The messages between node $u$ and its neighbors $\mathcal{N}_u$ are aggregated as $\widehat{\mathbf{m}}_u^{(l)} = \text{Agg}(\mathbf{m}_{uv}^{(l)} | v \in \mathcal{N}_u)$. (3) Update. Finally, GNN uses a non-linear function to update node embeddings as $\mathbf{h}_u^{(l)} = \text{Upd}(\widehat{\mathbf{m}}_u^{(l)}, \mathbf{h}_u^{(l-1)})$ using the aggregated message and the embedding from the previous layer. The obtained final node representation is then fed to a classifier or regressor to obtain a prediction $\widehat{\mu}(X)$.

**Conformal prediction.** In this work, we focus on the computationally efficient split conformal prediction method [43].[2] Given a predefined miscoverage rate $\alpha \in [0, 1]$, it proceeds in three steps: (1) non-conformity scores. CP first obtains any heuristic notion of uncertainty called non-conformity score $V : \mathcal{X} \times \mathcal{Y} \to \mathbb{R}$. Intuitively, $V(x, y)$ measures how $y$ "conforms" to the prediction at $x$. An example is the predicted probability of a class $y$ in classification or the residual value $V(x, y) = |y - \widehat{\mu}(x)|$ in regression for a predictor $\widehat{\mu} : \mathcal{X} \to \mathcal{Y}$. (2) Quantile computation. CP then takes the $1 - \alpha$ quantile of the non-conformity scores computed on the calibration set. Let $\{(X_i, Y_i)\}_{i=1}^n$ be the calibration data, where $n = |\mathcal{D}_{\text{calib}}|$, and compute $\widehat{\eta} = \text{quantile}(\{V(X_1, Y_1), \cdots, V(X_n, Y_n)\}, (1 - \alpha)(1 + \frac{1}{n}))$. (3) Prediction set/interval construction. Given a new test point $X_{n+1}$, CP constructs a prediction set/interval $C(X_{n+1}) = \{y \in \mathcal{Y} : V(X_{n+1}, y) \le \widehat{\eta}\}$. If $\{Z_i\}_{i=1}^{n+1} := \{(X_i, Y_i)\}_{i=1}^{n+1}$ are exchangeable,[3] then $V_{n+1} := V(X_{n+1}, Y_{n+1})$ is exchangeable with $\{V_i\}_{i=1}^n$ since $\widehat{\mu}$ is given. Thus, $\widehat{C}(X_{n+1})$ contains the true label with predefined coverage rate [43]: $P\{Y_{n+1} \in C(X_{n+1})\} = \mathbb{P}\{V_{n+1} \ge \text{Quantile}(\{V_1, \ldots, V_{n+1}\}, 1 - \alpha)\} \ge 1 - \alpha$ due to exchangeability of $\{V_i\}_{i=1}^{n+1}$. This framework works for any non-conformity score. CF-GNN is similarly non-conformity score-agnostic. However, for demonstration, we focus on two popular scores, described in detail below.

**Adaptive Prediction Set (APS).** For the classification task, we use the non-conformity score in APS proposed by [37]. It takes the cumulative sum of ordered class probabilities till the true class. Formally, given any estimator $\widehat{\mu}_j(x)$ for the conditional probability of $Y$ being class $j$ at $X = x$, $j = 1, \ldots, |\mathcal{Y}|$, we denote the cumulative probability till the $k$-th most promising class as $V(x, k) = \sum_{j=1}^k \widehat{\mu}_{\pi_{(j)}}(x)$,

---

[2]See Appendix B for discussion on full and split conformal prediction. We refer to the split conformal prediction when we describe conformal prediction throughout the paper.

[3]Exchangeability definition: for any $z_1, \ldots, z_{n+1}$ and any permutation $\pi$ of $\{1, \ldots, n+1\}$, it holds that $\mathbb{P}((Z_{\pi(1)}, \ldots, Z_{\pi(n+1)}) = (z_1, \ldots, z_{n+1})) = \mathbb{P}((Z_1, \ldots, Z_{n+1}) = (z_1, \ldots, z_{n+1}))$.

where $\pi$ is a permutation of $\mathcal{Y}$ so that $\widehat{\mu}_{\pi(1)}(x) \geq \widehat{\mu}_{\pi(2)}(x) \geq \cdots \geq \widehat{\mu}_{\pi(|\mathcal{Y}|)}(x)$. Then, the prediction set is constructed as $C(x) = \{\pi(1), \cdots, \pi(k^*)\}$, where $k^* = \inf\{k : \sum_{j=1}^{k} \widehat{\mu}_{\pi(j)}(x) \geq \widehat{\eta}\}$.

**Conformalized Quantile Regression (CQR).** For the regression task, we use CQR in [36]. CQR is based on quantile regression (QR). QR obtains heuristic estimates $\widehat{\mu}_{\alpha/2}(x)$ and $\widehat{\mu}_{1-\alpha/2}(x)$ for the $\alpha/2$-th and $1 - \alpha/2$-th conditional quantile functions of $Y$ given $X = x$. The non-conformity score is $V(x,y) = \max\{\widehat{\mu}_{\alpha/2}(x) - y, y - \widehat{\mu}_{1-\alpha/2}(x)\}$, interpreted as the residual of true label projected to the closest quantile. The prediction interval is then $C(x) = [\widehat{\mu}_{\alpha/2}(x) - \widehat{\eta}, \widehat{\mu}_{1-\alpha/2}(x) + \widehat{\eta}]$.

In its vanilla form, the non-conformity score (including APS and CQR) in CP does not depend on the calibration and test data. That means, $\{\mathbf{x}_v\}_{v \in \mathcal{D}_{\text{calib}} \cup \mathcal{D}_{\text{test}}}$ are not revealed in the training process of $V$, which is the key to exchangeability. In contrast, GNN training typically leverages the entire graph, and hence the learned model depends on the calibration and test attributes in a complicated way. In the following, for clarity, we denote any non-conformity score built on a GNN-trained model as

$$V(x, y; \{z_v\}_{v \in \mathcal{D}_{\text{train}} \cup \mathcal{D}_{\text{valid}}}, \{\mathbf{x}_v\}_{v \in \mathcal{D}_{\text{calib}} \cup \mathcal{D}_{\text{test}}}, \mathcal{V}, \mathcal{E})$$

to emphasize its dependence on the entire graph, where $z_v = (\mathbf{x}_v, Y_v)$ for $v \in \mathcal{V}$.

**Evaluation metrics.** The goal is to ensure valid marginal coverage while decreasing the inefficiency as much as possible. Given the test set $\mathcal{D}_{\text{test}}$, the empirical marginal coverage is defined as Coverage $:= \frac{1}{|\mathcal{D}_{\text{test}}|} \sum_{i \in \mathcal{D}_{\text{test}}} \mathbb{I}(Y_i \in C(X_i))$. For the regression task, inefficiency is measured as the interval length while for the classification task, the inefficiency is the size of the prediction set: Ineff $:= \frac{1}{|\mathcal{D}_{\text{test}}|} \sum_{i \in \mathcal{D}_{\text{test}}} |C(X_i)|$. The larger the length/size, the more inefficient. Note that inefficiency of conformal prediction is different from accuracy of the original predictions. Our method does not change the trained prediction but modifies the prediction sets from conformal prediction.

## 3 Exchangeability and Validity of Conformal Prediction on Graph

To deploy CP for graph-structured data, we first study the exchangeability of node information under the transductive setting. We show that under a general permutation invariant condition (Assumption 1), exchangeability of the non-conformity scores is still valid even though GNN training uses the calibration and test information; this paves the way for applying conformal prediction to GNN models. We develop an exact characterization of the test-time coverage of conformal prediction in such settings. Proofs of these results are in Appendix A.1.

**Assumption 1.** *For any permutation $\pi$ of $\mathcal{D}_{calib} \cup \mathcal{D}_{test}$, the non-conformity score $V$ obeys*

$$V(x, y; \{z_v\}_{v \in \mathcal{D}_{train} \cup \mathcal{D}_{valid}}, \{\mathbf{x}_v\}_{v \in \mathcal{D}_{calib} \cup \mathcal{D}_{test}}, \mathcal{V}, \mathcal{E})$$
$$= V(x, y; \{z_v\}_{v \in \mathcal{D}_{train} \cup \mathcal{D}_{valid}}, \{\mathbf{x}_{\pi(v)}\}_{v \in \mathcal{D}_{calib} \cup \mathcal{D}_{test}}, \mathcal{V}_\pi, \mathcal{E}_\pi),$$

*where $(\mathcal{V}_\pi, \mathcal{E}_\pi)$ represents a graph where $\mathcal{D}_{calib} \cup \mathcal{D}_{test}$ nodes (indices) are permuted according to $\pi$.*

Assumption 1 imposes a permutation invariance condition for the GNN training, i.e., model output/non-conformity score is invariant to permuting the ordering of the calibration and test nodes (with their edges permuted accordingly) on the graph. To put it differently, different selections of calibration sets do not modify the non-conformity scores for any node in the graph. GNN models (including those evaluated in our experiments) typically obey Assumption 1, because they only use the structures and attributes in the graph without information on the ordering of the nodes [24, 16, 12].

For clarity, we write the calibration data as $\{(X_i, Y_i)\}_{i=1}^{n}$, where $X_i = X_{v_i}$, and $v_i \in \mathcal{D}_{\text{calib}}$ is the $i$-th node in the calibration data under some pre-defined ordering. Similarly, the test data are $\{(X_{n+j}, Y_{n+j})\}_{j=1}^{m}$, where $X_{n+j} = X_{v_j}$, and $v_j \in \mathcal{D}_{\text{test}}$ is the $j$-th node in the test data. We write

$$V_i = V(X_i, Y_i; \{z_i\}_{i \in \mathcal{D}_{\text{train}} \cup \mathcal{D}_{\text{valid}}}, \{\mathbf{x}_v\}_{v \in \mathcal{D}_{\text{calib}} \cup \mathcal{D}_{\text{test}}}, \mathcal{V}, \mathcal{E}), \quad i = 1, \ldots, n, n+1, \ldots, n+m.$$

$V_i$ is a random variable that depends on the training process and the split of calibration and test data. The next lemma shows that under Assumption 1, the non-conformity scores are still exchangeable.

**Lemma 2.** *In the transductive setting described in Section 2, conditional on the entire unordered graph $(\mathcal{V}, \mathcal{E})$, all the attribute and label information $\{(\mathbf{x}_v, y_v)\}_{v \in \mathcal{V}}$, and the index sets $\mathcal{D}_{train}$ and $\mathcal{D}_{ct} := \mathcal{D}_{calib} \cup \mathcal{D}_{test}$, the unordered set of the scores $[V_i]_{i=1}^{n+m}$ are fixed. Also, the calibration scores $\{V_i\}_{i=1}^{n}$ are a simple random sample from $\{V_i\}_{i=1}^{n+m}$. That is, for any subset $\{v_1, \ldots, v_n\} \subseteq \{V_i\}_{i=1}^{n+m}$ of size $n$, $P(\{V_i\}_{i=1}^{n} = \{v_1, \ldots, v_n\} \mid \{V_i\}_{i=1}^{n+m}) = 1/\binom{n}{|\mathcal{D}_{ct}|}$.*

Based on Lemma 2, we next show that any permutation-invariant non-conformity score leads to valid prediction sets, and provide an exact characterization of the distribution of test-time coverage.

**Theorem 3.** *Given any score $V$ obeying Assumption 1 and any confidence level $\alpha \in (0, 1)$, we define the split conformal prediction set as $\widehat{C}(x) = \{y \colon V(x, y) \leq \widehat{\eta}\}$, where*

$$\widehat{\eta} = \inf \left\{ \eta \colon \tfrac{1}{n} \sum_{i=1}^{n} \mathbb{1}\{V(X_i, Y_i) \leq \eta\} \geq (1 - \alpha)(1 + 1/n) \right\}.$$

*Then $\mathbb{P}(Y_{n+j} \in \widehat{C}(X_{n+j})) \geq 1 - \alpha, \forall j = 1, \ldots, m$. Moreover, define $\widehat{\text{Cover}} = \tfrac{1}{m} \sum_{j=1}^{m} \mathbb{1}\{Y_{n+j} \in \widehat{C}(X_{n+j})\}$. If the $V_i$'s, $i \in \mathcal{D}_{calib} \cup \mathcal{D}_{test}$, have no ties almost surely, then for any $t \in (0, 1)$,*

$$\mathbb{P}(\widehat{\text{Cover}} \leq t) = 1 - \Phi_{\mathrm{HG}}\big(\lceil (n+1)(1-q) \rceil - 1; m+n, n, \lceil (1-q)(n+1) \rceil + \lceil mt \rceil\big),$$

*where $\Phi_{\mathrm{HG}}(\cdot; N, n, k)$ denotes the cumulative distribution function of a hyper-geometric distribution with parameters $N, n, k$ (drawing $k$ balls from an urn wherein $n$ out of $N$ balls are white balls).*

Figure 2 plots the probability density functions (p.d.f.) of $\widehat{\text{Cover}}$ at a sequence of $t \in [0, 1]$ fixing $n = 1000$ while varying $m$. The exact distribution described in Theorem 3 is useful in determining the size of the calibration data in order for the test-time coverage to concentrate sufficiently tightly around $1 - \alpha$. More discussion and visualization of $\widehat{\text{Cover}}$ under different values of $(n, m)$ are in Appendix A.2. Note that similar exchangeability and validity results are obtained in several concurrent works [15, 33], yet without exact characterization of the test-time coverage.

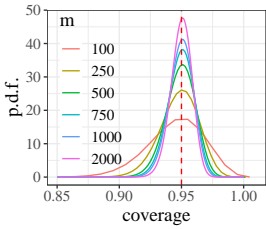

**Figure 2:** P.d.f. of $\widehat{\text{Cover}}$ for $n = 1000$ and $\alpha = 0.05$; curves represent different values of test sample size $m$.

## 4 CF-GNN: Conformalized Graph Neural Networks

We now propose a new method called CF-GNN to reduce inefficiency while maintaining valid coverage. The key idea of CF-GNN is to boost any given non-conformity score with graphical information. The method illustration is in Figure 1 and pseudo-code is in Appendix C.

**Efficiency-aware boosting.** Standard CP takes any pre-trained predictor to construct the prediction set/interval (see Section 2). A key observation is that the training stage is not aware of the post-hoc stage of conformal prediction set/interval construction. Thus, it is not optimized for efficiency. Our high-level idea is to include an additional correction step that boosts the non-conformity score, which happens after the model training and before conformal prediction. To ensure flexibility and practicality, our framework is designed as a generic wrapper that works for any pre-trained GNN model, without changing the base model training process.

**Motivation: Inefficiency correlation.** Our approach to boosting the scores is based on exploiting the correlation among connected nodes. Since the connected nodes usually represent entities that interact in the real world, there can be strong correlation between them. To be more specific, it is well established in network science that prediction residuals are correlated along edges [20]. Such a result implies a similar phenomenon for inefficiency: taking CQR for regression as an example, the prediction interval largely depends on the residual of the true outcome from the predicted quantiles. Hence, the prediction interval lengths are also highly correlated for connected nodes. We empirically verify this intuition in Figure 3, where we plot the difference in the prediction interval lengths for connected/unconnected node pairs in the Anaheim dataset using vanilla CQR for GCN.[4] In Figure 3, we observe that inefficiency has a topological root: connected nodes usually have similar residual scales, suggesting the existence of rich neighborhood information for the residuals. This motivates us to utilize such information to correct the scores and achieve better efficiency.

**Topology-aware correction model.** Based on the motivation above, we update the model predictions using the neighbor predictions. However, the relationship in neighborhood predictions could be complex (i.e. beyond homophily), making heuristic aggregation such as averaging/summing/etc. overly simplistic and not generalizable to all types of graphs. Ideally, we want to design a general and powerful mechanism that flexibly aggregates graph information. This requirement could be perfectly

---

[4]That is, we take the predicted quantiles from GCN and directly builds prediction intervals using CQR.

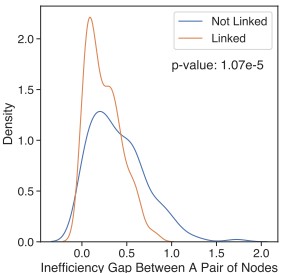

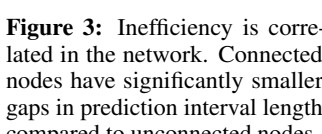

**Figure 3:** Inefficiency is correlated in the network. Connected nodes have significantly smaller gaps in prediction interval length compared to unconnected nodes.

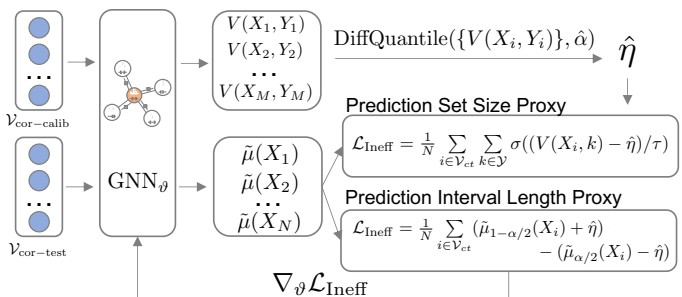

**Figure 4:** We simulate the downstream conformal step and optimize for inefficiency directly. We first produce differentiable quantile $\widehat{\eta}$ using $V(X_i, Y_i)$ from $\mathcal{V}_{\text{cor-cal}}$. We then construct a prediction set size/interval length proxy on $\mathcal{V}_{\text{cor-test}}$ and directly minimize inefficiency loss by updating $\text{GNN}_\vartheta$.

fulfilled by GNN message passing as it represents a class of learnable aggregation functions over graphs. Therefore, we use a separate GNN learner parameterized by $\vartheta$ for the same network $G$ but with modified input node features; specifically, we use the base GNN prediction ($\mathbf{X}_0 = \widehat{\mu}(X)$) as input, and output $\tilde{\mu}(X) = \text{GNN}_\vartheta(\widehat{\mu}(X), G)$. We will then use $\tilde{\mu}(X)$ as the input for constructing conformal prediction sets/intervals. Note that this second GNN model is a post-hoc process and only requires base GNN predictions, instead of access to the base GNN model.

**Training with conformal-aware inefficiency loss.** Given the hypothesis class, it remains to devise a concrete recipe for training the correction model $\text{GNN}_\vartheta$ parameters. Recall that as with many other prediction models, a GNN model is typically trained to optimize prediction loss (i.e.cross-entropy loss or mean squared error) but not geared towards efficiency for the post-hoc conformal prediction step. We design $\text{GNN}_\vartheta$ to be efficiency-aware by proposing a differentiable inefficiency loss that $\text{GNN}_\vartheta$ optimizes over; this allows integration of GNN message passing to exploit the neighborhood information and also ensures a good $\tilde{\mu}(\cdot)$ that leads to efficient prediction sets in downstream steps.

We first withhold a small fraction $\gamma$ of the calibration dataset and use it for the correction procedure. The remaining data is used as the "usual" calibration data for building the conformal prediction set. We then further split the withheld data into a correction calibration set $\mathcal{V}_{\text{cor-cal}}$ and correction testing set $\mathcal{V}_{\text{cor-test}}$, to simulate the downstream conformal inference step. Given $\tilde{\mu}(X) = \text{GNN}_\vartheta(\widehat{\mu}(X), G)$ and a target miscoverage rate $\alpha$, the framework follows three steps:

(1) Differentiable quantile: we compute a smooth quantile $\widehat{\eta}$ based on the $\mathcal{V}_{\text{cor-cal}}$ by

$$\widehat{\eta} = \text{DiffQuantile}(\{V(X_i, Y_i)|i \in \mathcal{V}_{\text{cor-cal}}\}, (1-\alpha)(1 + 1/|\mathcal{V}_{\text{cor-cal}}|)).$$

Since the non-conformity score is usually differentiable, it only requires differentiable quantile calculation where there are well-established methods available [6, 5].

(2) Differentiable inefficiency proxy: we then construct a differentiable proxy $\mathbf{c}$ of the inefficiency on $\mathcal{V}_{\text{cor-test}}$ by simulating the downstream conformal prediction procedures. We propose general formulas to construct $\mathbf{c}$ that are applicable for any classification and regression tasks respectively:

*a. Inefficiency loss instantiation for Classification*: The desirable proxy is to simulate the prediction set size using $\mathcal{D}_{\text{cor-test}}$ as the "test" data and $\mathcal{D}_{\text{cor-calib}}$ as the "calibration" data. For class $k$ and node $i$ in $\mathcal{D}_{\text{cor-test}}$, the non-conformity score is $V(X_i, k)$ for class $k$, where $V(\cdot)$, for instance, is the APS score in Section 2. Then, we define the inefficiency proxy as

$$\mathbf{c}_{i,k} = \sigma(\frac{V(X_i, k) - \widehat{\eta}}{\tau}),$$

where $\sigma(x) = \frac{1}{1+e^{-x}}$ is the sigmoid function and $\tau$ is a temperature hyper-parameter [40]. It can be interpreted as a soft assignment of class $k$ to the prediction set. When $\tau \to 0$, it becomes a hard assignment. The per-sample inefficiency proxy is then readily constructed as $\mathbf{c}_i = \frac{1}{|\mathcal{Y}|} \sum_{k \in \mathcal{Y}} \mathbf{c}_{i,k}$.

*b. Inefficiency loss instantiation for Regression*: The desirable proxy is to simulate the prediction interval length. For node $i$ in $\mathcal{V}_{\text{cor-test}}$, the conformal prediction interval is $[\tilde{\mu}_{\alpha/2}(X_i) - \widehat{\eta}, \tilde{\mu}_{1-\alpha/2}(X_i) + \widehat{\eta}]$. Thus, the per-sample prediction interval length could be directly calculated as

$$\mathbf{c}_i = (\tilde{\mu}_{1-\alpha/2}(X_i) + \widehat{\eta}) - (\tilde{\mu}_{\alpha/2}(X_i) - \widehat{\eta}).$$

Since $\text{GNN}_\vartheta$ maps intervals to intervals and do not pose a constraint on the prediction, it may incur a trivial optimized solution where $\tilde{\mu}_{1-\alpha/2}(X) < \tilde{\mu}_{\alpha/2}(X)$. Thus, we pose an additional consistency regularization term: $(\tilde{\mu}_{1-\alpha/2}(X) - \widehat{\mu}_{1-\alpha/2}(X))^2 + (\tilde{\mu}_{\alpha/2}(X) - \widehat{\mu}_{\alpha/2}(X))^2$. This regularizes the updated intervals to not deviate significantly to reach the trivial solution.

(3) Inefficiency loss: finally, the inefficiency loss is an average of inefficiency proxies $L_{\text{ineff}} = \frac{1}{|\mathcal{V}_{\text{cor}-\text{test}}|} \sum_i \mathbf{c}_i$. The $\text{GNN}_\vartheta$ is optimized using backpropagation in an end-to-end fashion.

**Conditional coverage.** A natural question is whether optimizing the efficiency of conformal prediction may hurt its conditional validity.[5] In Section 5, we empirically demonstrate satisfactory conditional coverage across various graph features, which even improves upon the direct application of APS and CQR to graph data. We conjecture that it is because we correct for the correlation among nodes. However, theoretical understanding is left for future investigation.

**Graph exchangeability.** The post-hoc correction model is GNN-based, thus, it is permutation-invariant. Thus, it satisfies the exchangeability condition laid out in our theory in Section 3. Empirically, we demonstrate in Section 5 that CF-GNN achieves target empirical marginal coverage.

**Computational cost.** We remark that CF-GNN scales similarly as base GNN training since the correction step follows a standard GNN training procedure but with a modified input attribute and loss function. Notably, as the input to the correction model usually has a smaller attribute size (the number of classes for classification and 2 for regression), it has smaller parameter size than standard GNN training. In addition, it is also compatible with standard GNN mini-batching techniques.

**General loss functions.** Finally, we note that the choice of our loss function can be quite general. For instance, one may directly optimize for conditional validity by choosing a proper loss function.

## 5  Experiments

We conduct experiments to demonstrate the advantages of CF-GNN over other UQ methods in achieving empirical marginal coverage for graph data, as well as the efficiency improvement with CF-GNN. We also evaluate conditional coverage of CF-GNN and conduct systematic ablation and parameter analysis to show the robustness of CF-GNN.

**Evaluation setup.** For node classification, we follow a standard semi-supervised learning evaluation procedure [24], where we randomly split data into folds with 20%/10%/70% nodes as $\mathcal{D}_{\text{train}}/\mathcal{D}_{\text{valid}}/\mathcal{D}_{\text{calib}} \cup \mathcal{D}_{\text{test}}$. For the node regression task, we follow a previous evaluation procedure from [20] and randomly split the data into folds with 50%/10%/40% nodes as $\mathcal{D}_{\text{train}}/\mathcal{D}_{\text{valid}}/\mathcal{D}_{\text{calib}} \cup \mathcal{D}_{\text{test}}$. We conduct 100 random splits of calibration/testing sets to estimate the empirical coverage. Using the test-time coverage distribution in Figure 2 to ensure that coverage is concentrated tightly around $1$-$\alpha$, we modify the calibration set size to $\min\{1000, |\mathcal{D}_{\text{calib}} \cup \mathcal{D}_{\text{test}}|/2\}$, and use the rest as the test sample. For a fair comparison, we first train 10 runs of the base GNN model and then fix the predictions (i.e. the input to UQ baselines and CF-GNN). In this way, we ensure that the gain is not from randomness in base model training. The hyperparameter search strategy and configurations for CF-GNN and baselines can be found in Appendix D.1.

**Models & baselines to evaluate coverage.** For classification, we first use general statistical calibration approaches including temperate scaling [13], vector scaling [13], ensemble temperate scaling [49]. We also use SOTA GNN-specific calibration learners including CaGCN [44] and GATS [17]. The prediction set is the set of classes from highest to lowest scores until accumulative scores exceed $1$-$\alpha$. For regression, we construct prediction intervals using quantile regression (QR) [25], Monte Carlo dropouts (MC dropout) [9], and Bayesian loss to model both aleatoric and epistemic uncertainty [23]. More information about baselines can be found in Appendix D.2.

**Models & baselines to evaluate efficiency.** As smaller coverage always leads to higher efficiency, for a fair comparison, we can only compare methods on efficiency that achieve the same coverage. Thus, we do not evaluate UQ baselines here since they do not produce exact coverage and are thus not comparable. While any CP-based methods produce exact coverage, to the best of our knowledge,

---

[5]Conditional coverage asks for $\mathbb{P}(Y_{n+j} \in \widehat{C}(X_{n+j}) \,|\, X_{n+j} = x) \approx 1 - \alpha$ for all $x \in \mathcal{X}$. Although exact conditional validity is statistically impossible [8], approximate conditional validity is a practically important property that APS and CQR are designed for. See Section 5 for common ways to assess conditional coverage.

**Table 1:** Empirical marginal coverage of node classification(upper table) and node regression tasks(lower table). The result takes the average and standard deviation across 10 GNN runs with 100 calib/test splits. ✔ means that the UQ method reaches the target coverage (i.e. coverage $\geq 0.95$) while ✗ means that it fails to reach it. The last column "Covered" becomes ✔ if a UQ method reaches target coverage for all datasets and ✗ vice versus.

| Task | UQ Model | Cora | DBLP | CiteSeer | PubMed | Computers | Photo | CS | Physics | Covered? |
|---|---|---|---|---|---|---|---|---|---|---|
| Node classif. | Temp. Scale. | 0.946±.003 ✗ | 0.920±.009 ✗ | 0.952±.004 ✔ | 0.899±.002 ✗ | 0.929±.002 ✗ | 0.962±.002 ✔ | 0.957±.001 ✔ | 0.969±.000 ✔ | ✗ |
| | Vector Scale. | 0.944±.004 ✗ | 0.921±.009 ✗ | 0.951±.004 ✔ | 0.899±.003 ✗ | 0.932±.002 ✗ | 0.963±.002 ✔ | 0.958±.001 ✔ | 0.969±.000 ✔ | ✗ |
| | Ensemble TS | 0.947±.003 ✗ | 0.920±.008 ✗ | 0.953±.003 ✔ | 0.899±.002 ✗ | 0.930±.002 ✗ | 0.964±.002 ✔ | 0.958±.001 ✔ | 0.969±.000 ✔ | ✗ |
| | CaGCN | 0.939±.005 ✗ | 0.922±.004 ✗ | 0.949±.005 ✗ | 0.898±.003 ✗ | 0.926±.003 ✗ | 0.956±.002 ✔ | 0.954±.003 ✔ | 0.968±.001 ✔ | ✗ |
| | GATS | 0.939±.005 ✗ | 0.921±.004 ✗ | 0.951±.005 ✔ | 0.898±.002 ✗ | 0.925±.002 ✗ | 0.957±.002 ✔ | 0.957±.001 ✔ | 0.968±.000 ✔ | ✗ |
| | CF-GNN | 0.952±.001 ✔ | 0.952±.001 ✔ | 0.953±.001 ✔ | 0.953±.001 ✔ | 0.952±.001 ✔ | 0.953±.001 ✔ | 0.952±.001 ✔ | 0.952±.001 ✔ | ✔ |

| Task | UQ Model | Anaheim | Chicago | Education | Election | Income | Unemploy. | Twitch | Covered? |
|---|---|---|---|---|---|---|---|---|---|
| Node regress. | QR | 0.943±.031 ✗ | 0.950±.007 ✗ | 0.959±.001 ✔ | 0.956±.004 ✔ | 0.960±.005 ✔ | 0.954±.004 ✔ | 0.900±.015 ✗ | ✗ |
| | MC dropout | 0.553±.022 ✗ | 0.427±.015 ✗ | 0.423±.013 ✗ | 0.417±.008 ✗ | 0.532±.022 ✗ | 0.489±.016 ✗ | 0.448±.017 ✗ | ✗ |
| | BayesianNN | 0.967±.001 ✔ | 0.955±.003 ✔ | 0.957±.002 ✔ | 0.958±.009 ✔ | 0.970±.004 ✔ | 0.960±.001 ✔ | 0.923±.006 ✗ | ✗ |
| | CF-GNN | 0.957±.003 ✔ | 0.954±.002 ✔ | 0.951±.001 ✔ | 0.950±.001 ✔ | 0.951±.001 ✔ | 0.951±.001 ✔ | 0.954±.001 ✔ | ✔ |

there are no existing graph-based conformal prediction methods for transductive settings. Thus, we can only compare with the direct application of conformal prediction (CP) to base GNN. In the main text, we only show results for GCN [24] as the base model; results of three additional popular GNN models (GraphSAGE [16], GAT [42], and SGC [45]) are deferred to Appendix D.4.

**Datasets.** We evaluate CF-GNN on 8 node classification datasets and 7 node regression datasets with diverse network types such as geographical network, transportation network, social network, citation network, and financial network. Dataset statistics are in Appendix D.3.

## 5.1 Results

**CF-GNN achieves empirical marginal coverage while existing UQ methods do not.** We report marginal coverage of various UQ methods with target coverage at 95% (Table 1). There are three key takeaways. Firstly, none of these UQ methods achieves the target coverage for all datasets while CF-GNN does, highlighting the lack of statistical rigor in those methods and the necessity for a guaranteed UQ method. Secondly, it validates our theory from Section 3 that CF-GNN achieves designated coverage in transductive GNN predictions. Lastly, CF-GNN achieves empirical coverage that is close to the target coverage while baseline UQ methods are not. This controllable feature of CF-GNN is practically useful for practitioners that aim for a specified coverage in settings such as planning and selection.

**CF-GNN significantly reduces inefficiency.** We report empirical inefficiency for 8 classification and 7 regression datasets (Table 2). We observe that we achieve consistent improvement across datasets with up to 74% reduction in the prediction set size/interval length. We additionally conduct the same experiments for 3 other GNN models including GAT, GraphSAGE, and SGC in Appendix D.4 and we observe that performance gain is generalizable to diverse architecture choices. Furthermore, CF-GNN yields more efficient prediction sets than existing UQ methods even if we manually adjust the nominal level of them to achieve 95% empirical coverage (it is however impossible to do so in practice, here we do this for evaluation). For instance, the best calibration method GATS yields an average prediction size of 1.82 on Cora when the nominal level is tuned to achieve 95% empirical coverage. In contrast, CF-GNN has an average size of 1.76, smaller than GATS. In Appendix D.5, we also observe that CF-GNN also reduces inefficiency for advanced conformal predictor RAPS for classification task. In addition, we find that CF-GNN yields little changes to the prediction accuracy of the original GNN model (Appendix D.7).

**CF-GNN empirically maintains conditional coverage.** While CF-GNN achieves marginal coverage, it is highly desirable to have a method that achieves reasonable conditional coverage, which was the motivation of APS and CQR. We follow [37] to evaluate conditional coverage via the Worst-Slice (WS) coverage, which takes the worst coverage across slices in the feature space (i.e. node input features). We observe that CF-GNN achieves a WS coverage close to $1 - \alpha$, indicating satisfactory conditional coverage (Cond. Cov. (Input Feat.) row in Table 3). Besides the raw features, for each node, we also construct several network features (which are label agnostic) including clustering

**Table 2:** Empirical inefficiency measured by the size/length of the prediction set/interval for node classification (left table)/regression(right table). A smaller number has better efficiency. We show the relative improvement (%) of CF-GNN over CP on top of the →. The result uses APS for classification and CQR for regression with GCN as the base model. Additional results on other GNN models are at Appendix D.4. We report the average and standard deviation of prediction sizes/lengths calculated from 10 GNN runs, each with 100 calibration/test splits.

| Task | Dataset | CP $\xrightarrow{}$ CF-GNN |
|------|---------|------------|
| Node classif. | Cora | $3.80_{\pm.28} \xrightarrow{-53.61\%} 1.76_{\pm.27}$ |
| | DBLP | $2.43_{\pm.03} \xrightarrow{-49.13\%} 1.23_{\pm.01}$ |
| | CiteSeer | $3.86_{\pm.11} \xrightarrow{-74.27\%} 0.99_{\pm.02}$ |
| | PubMed | $1.60_{\pm.02} \xrightarrow{-19.05\%} 1.29_{\pm.03}$ |
| | Computers | $3.56_{\pm.13} \xrightarrow{-49.05\%} 1.81_{\pm.12}$ |
| | Photo | $3.79_{\pm.13} \xrightarrow{-56.28\%} 1.66_{\pm.21}$ |
| | CS | $7.79_{\pm.29} \xrightarrow{-62.16\%} 2.95_{\pm.49}$ |
| | Physics | $3.11_{\pm.07} \xrightarrow{-62.81\%} 1.16_{\pm.13}$ |
| Average Improvement | | -53.75% |

| Task | Dataset | CP $\xrightarrow{}$ CF-GNN |
|------|---------|------------|
| Node regress. | Anaheim | $2.89_{\pm.39} \xrightarrow{-25.00\%} 2.17_{\pm.11}$ |
| | Chicago | $2.05_{\pm.07} \xrightarrow{-0.48\%} 2.04_{\pm.17}$ |
| | Education | $2.56_{\pm.02} \xrightarrow{-5.07\%} 2.43_{\pm.05}$ |
| | Election | $0.90_{\pm.01} \xrightarrow{+0.21\%} 0.90_{\pm.02}$ |
| | Income | $2.51_{\pm.12} \xrightarrow{-4.58\%} 2.40_{\pm.05}$ |
| | Unemploy | $2.72_{\pm.03} \xrightarrow{-10.83\%} 2.43_{\pm.04}$ |
| | Twitch | $2.43_{\pm.10} \xrightarrow{-1.36\%} 2.39_{\pm.07}$ |
| Average Improvement | | -6.73% |

**Table 3:** CF-GNN achieves conditional coverage, measured by Worse-Slice Coverage [37]. We use Cora/Twitch as an example classification/regression dataset. Results on other network features and results on target coverage of 0.9 can be found in Appendix D.6.

| Target: 0.95 | Classification | | Regression | |
|--------------|------|--------|------|--------|
| Model | CP | CF-GNN | CP | CF-GNN |
| Marginal Cov. | $0.95_{\pm.01}$ | $0.95_{\pm.01}$ | $0.96_{\pm.02}$ | $0.96_{\pm.02}$ |
| Cond. Cov. (Input Feat.) | $0.94_{\pm.02}$ | $0.94_{\pm.03}$ | $0.95_{\pm.04}$ | $0.94_{\pm.05}$ |
| Cond. Cov. (Cluster) | $0.89_{\pm.06}$ | $0.93_{\pm.04}$ | $0.96_{\pm.03}$ | $0.96_{\pm.03}$ |
| Cond. Cov. (Between) | $0.81_{\pm.06}$ | $0.95_{\pm.03}$ | $0.94_{\pm.05}$ | $0.94_{\pm.05}$ |
| Cond. Cov. (PageRank) | $0.78_{\pm.06}$ | $0.94_{\pm.03}$ | $0.94_{\pm.05}$ | $0.94_{\pm.05}$ |

**Table 4:** Ablation. For Size/length, we use Cora/Anaheim dataset with GCN backbone. Each experiment is with 10 independent base model runs with 100 conformal split runs.

| Topology -aware | Ineff. Loss | Size | Length |
|----------|-------|------|--------|
| ✔ | ✔ | $1.76_{\pm.27}$ | $2.17_{\pm.11}$ |
| ✔ | ✗ | $2.42_{\pm.35}$ | $2.23_{\pm.10}$ |
| ✗ | ✔ | $2.35_{\pm.47}$ | $2.32_{\pm.18}$ |
| ✗ | ✗ | $3.80_{\pm.28}$ | $2.89_{\pm.39}$ |

coefficients, betweenness, PageRank, closeness, load, and harmonic centrality, and then calculate the WS coverage over the network feature space. We observe close to 95% WS coverage for various network features, suggesting CF-GNN also achieves robust conditional coverage over network properties. We also see that the direct application of CP (i.e. without graph correction) has much smaller WS coverage for classification, suggesting that adjusting for neigborhood information in CF-GNN implicitly improves conditional coverage.

**Ablation.** We conduct ablations in Table 4 to test two main components in CF-GNN, topology-aware correction model, and inefficiency loss. We first remove the inefficiency loss and replace it with standard prediction loss. The performance drops as expected, showing the power of directly modeling inefficiency loss in the correction step. Secondly, we replace the GNN correction model with an MLP correction model. The performance drops significantly, showing the importance of the design choice of correction model and justifying our motivation on inefficiency correlation over networks.

**Parameter analysis.** We conduct additional parameter analysis to test the robustness of CF-GNN. We first adjust the target coverage rate and calculate the inefficiency (Figure 5(1)). CF-GNN consistently beats the vanilla CP across all target coverages. Moreover, we adjust the fraction $\gamma$ of the holdout calibration data in building the inefficiency loss, and observe that CF-GNN achieves consistent improvement in inefficiency (Figure 5(2)). We also observe a small fraction (10%) leads to excellent performance, showing that our model only requires a small amount of data for the inefficiency loss and leaves the majority of the calibration data for downstream conformal prediction.

## 6 Related Works

We discuss here related works that are closest to the ideas in CF-GNN and provide extended discussion on other related works in Appendix E.

(1) Uncertainty quantification (UQ) for GNN: Many UQ methods have been proposed to construct model-agnostic uncertain estimates for both classification [13, 49, 14, 27, 1] and regression [25, 41,

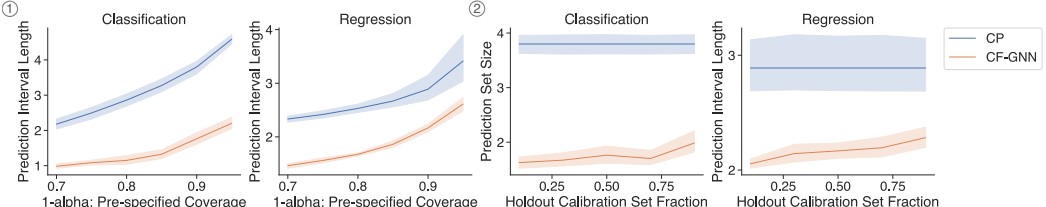

**Figure 5:** (1) Parameter analysis on inefficiency given different target coverage rate $1-\alpha$. (2) Parameter analysis on inefficiency given calibration set holdout fraction. Analyses use Cora/Anaheim for classification/regression.

38, 9, 28, 26, 35, 23, 19]. Recently, specialized calibration methods for GNNs that leverage network principles such as homophily have been developed [44, 17]. However, these UQ methods can fail to provide a statistically rigorous and empirically valid coverage guarantee (see Table 1). In contrast, CF-GNN achieves valid marginal coverage in both theory and practice.

(2) Conformal prediction for GNN: The application of CP to graph-structured data remains largely unexplored. [7] claims that nodes in the graph are not exchangeable in the inductive setting and employs the framework of [3] to construct prediction sets using neighborhood nodes as the calibration data for mitigating the miscoverage due to non-exchangeability. In contrast, we study the transductive setting where certain exchangeability property holds; thus, the method from [3] are not comparable to ours. Concurrent with our work, [15] studies the exchangeability under transductive setting and proposes a diffusion-based method for improving efficiency, which can be viewed as an instantiation of our approach where the GNN correction learns an identity mapping; [33] studies exchangeability in network regression for of non-conformity scores based on various network structures, with similar observations as our Theorem 3. Other recent efforts in conformal prediction for graphs include [32, 34] which focus on distinct problem settings.

(3) Efficiency of conformal prediction: How to achieve desirable properties beyond validity is an active topic in the CP community; we focus on the efficiency aspect here. One line of work designs "good" non-conformity scores in theory such as APS [37] and CQR [36]. More recent works take another approach, by modifying the training process of the prediction model. CF-GNN falls into the latter case, although our idea applies to any non-conformity score. ConfTr [40] also modifies training for improved efficiency. Our approach differs from theirs in significant ways. First, we develop a theory on CP validity on the graph data and leverage topological principles that are specialized to graph to improve efficiency while ConfTr focuses on i.i.d. vision image data. Also, ConfTr happens during base model training using the training set, while CF-GNN conducts post-hoc correction using withheld calibration data without assuming access to base model training, making ConfTr not comparable to us. Finally, we also propose a novel loss for efficiency in regression tasks.

## 7  Conclusion

In this work, we extend conformal prediction to GNNs by laying out the theoretical conditions for finite-sample validity and proposing a flexible graph-based CP framework to improve efficiency. Potential directions for future work include generalizing the inefficiency loss to other desirable CP properties such as robustness and conditional coverage; extensions to inductive settings or transductive but non-random split settings; extensions to other graph tasks such as link prediction, community detection, and so on.

## 8  Acknowledgements

K.H. and J.L. gratefully acknowledge the support of DARPA under Nos. HR00112190039 (TAMI), N660011924033 (MCS); ARO under Nos. W911NF-16-1-0342 (MURI), W911NF-16-1-0171 (DURIP); NSF under Nos. OAC-1835598 (CINES), OAC-1934578 (HDR), CCF-1918940 (Expeditions), NIH under No. 3U54HG010426-04S1 (HuBMAP), Stanford Data Science Initiative, Wu Tsai Neurosciences Institute, Amazon, Docomo, GSK, Hitachi, Intel, JPMorgan Chase, Juniper Networks, KDDI, NEC, and Toshiba. The content is solely the responsibility of the authors and does not necessarily represent the official views of the funding entities.

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

# A   Deferred details for Section 3

## A.1   Technical proofs for theoretical results

*Proof of Lemma 2.* Hereafter, we condition on the entire unordered graph, all the attribute and label information, and the index sets $\mathcal{D}_{\text{train}}$ and $\mathcal{D}_{\text{ct}}$. We define the scores evaluated at the original node indices as

$$v_v = V(\mathbf{x}_v, y_v; \{z_i\}_{i \in \mathcal{D}_{\text{train}} \cup \mathcal{D}_{\text{valid}}}, \{\mathbf{x}_v\}_{v \in \mathcal{D}_{\text{calib}} \cup \mathcal{D}_{\text{test}}}, \mathcal{V}, \mathcal{E}), \quad v \in \mathcal{D}_{\text{calib}} \cup \mathcal{D}_{\text{test}} \subseteq \mathcal{V}.$$

By Assumption 1, for any permutation $\pi$ of $\mathcal{D}_{\text{ct}}$, we always have

$$v_v = V(\mathbf{x}_v, y_v; \{z_i\}_{i \in \mathcal{D}_{\text{train}} \cup \mathcal{D}_{\text{valid}}}, \{\mathbf{x}_{\pi(v)}\}_{v \in \mathcal{D}_{\text{calib}} \cup \mathcal{D}_{\text{test}}}, \mathcal{V}_\pi, \mathcal{E}_\pi).$$

That is, given $\mathcal{D}_{\text{ct}}$, the evaluated score at any $v \in \mathcal{D}_{\text{ct}}$ remains invariant no matter which subset of $\mathcal{D}_{\text{ct}}$ is designated as $\mathcal{D}_{\text{calib}}$. This implies that the scores are fixed after conditioning:

$$[V_1, \ldots, V_{n+m}] = [v_v]_{v \in \mathcal{D}_{\text{ct}}},$$

where we use $[]$ to emphasize unordered sets. Thus, the calibration scores $\{V_i\}_{i=1}^n$ is a subset of size $n$ of $[v_v]_{v \in \mathcal{D}_{\text{ct}}}$. Note that under random splitting in the transductive setting, any permutation $\pi$ of $\mathcal{D}_{\text{ct}}$ occurs with the same probability, which gives the conditional probability in Lemma 2. □

*Proof of Theorem 3.* Throughout this proof, we condition on the entire unordered graph, all the attribute and label information, and the index sets $\mathcal{D}_{\text{train}}$ and $\mathcal{D}_{\text{ct}}$. By Lemma 2, after the conditioning, the unordered set of $\{V_i\}_{i=1}^{n+m}$ is fixed as $[v_v]_{v \in \mathcal{D}_{\text{ct}}}$, and $\{V_i\}_{i=1}^n$ is a simple random sample from $[v_v]_{v \in \mathcal{D}_{\text{ct}}}$. As a result, any test sample $V(X_{n+j}, Y_{n+j}), j = 1, \ldots, m$ is exchangeable with $\{V_i\}_{i=1}^n$. By standard theory for conformal prediction [43], this ensures *valid marginal coverage*, i.e., $\mathbb{P}(Y_{n+j} \in \widehat{C}(X_{n+j})) \geq 1 - \alpha$, where the expectation is over all the randomness.

We now proceed to analyze the distribution of $\widehat{\text{Cover}}$. For notational convenience, we write $N = m + n$, and view $\mathcal{D}_{\text{ct}}$ as the 'population'. In this way, $\{V_i\}_{i=1}^n$ is a simple random sample from $[v_v]_{v \in \mathcal{D}_{\text{ct}}}$. For every $\eta \in \mathbb{R}$, we define the 'population' cumulative distribution function (c.d.f.)

$$F(\eta) = \frac{1}{N} \sum_{v \in \mathcal{D}_{\text{ct}}} \mathbb{1}\{v_v \leq \eta\},$$

which is a deterministic function. We also define the calibration c.d.f. as

$$\widehat{F}_n(\eta) = \frac{1}{n} \sum_{v \in \mathcal{D}_{\text{calib}}} \mathbb{1}\{v_v \leq \eta\} = \frac{1}{n} \sum_{i=1}^n \mathbb{1}\{V_i \leq \eta\},$$

which is random, and its randomness comes from which subset of $\mathcal{D}_{\text{ct}}$ is $\mathcal{D}_{\text{calib}}$. By definition,

$$\widehat{\eta} = \inf\{\eta \colon \widehat{F}_n(\eta) \geq (1-q)(1+1/n)\}.$$

Since the scores have no ties, we know

$$\widehat{F}_n(\widehat{\eta}) = \lceil (1-q)(n+1) \rceil / n.$$

The test-time coverage can be written as

$$\begin{aligned}
\widehat{\text{Cover}} &= \frac{1}{m} \sum_{j=1}^m \mathbb{1}\{V_{n+j} \leq \widehat{\eta}\} \\
&= \frac{1}{N-n}\left( \sum_{v \in \mathcal{D}_{\text{ct}}} \mathbb{1}\{v_v \leq \widehat{\eta}\} - \sum_{v \in \mathcal{D}_{\text{calib}}} \mathbb{1}\{v_v \leq \widehat{\eta}\} \right) \\
&= \frac{N}{N-n} F(\widehat{\eta}) - \frac{n}{N-n} \widehat{F}_n(\widehat{\eta}) = \frac{N}{N-n} F(\widehat{\eta}) - \frac{\lceil (1-q)(n+1) \rceil}{N-n}.
\end{aligned}$$

Now we characterize the distribution of $\widehat{\eta}$. For any $\eta \in \mathbb{R}$, by the definition of $\widehat{\eta}$,

$$\mathbb{P}(\widehat{\eta} \leq \eta) = \mathbb{P}\big(n\widehat{F}_n(\eta) \geq (n+1)(1-q)\big) = \mathbb{P}\big(n\widehat{F}_n(\eta) \geq \lceil (n+1)(1-q) \rceil\big).$$

Note that $n\widehat{F}_n(\eta) = \sum_{v \in \mathcal{D}_{\text{calib}}} \mathbb{1}\{v_v \leq \eta\}$ is the count of data in $\mathcal{D}_{\text{calib}}$ such that the score is below $\eta$. By the simple random sample (i.e., sampling without replacement), $n\widehat{F}_n(\eta)$ follows a hyper-geometric distribution with parameter $N, n, NF(\eta)$. That is,

$$\mathbb{P}(n\widehat{F}_n(\eta) = k) = \frac{\binom{NF(\eta)}{k}\binom{N-NF(\eta)}{n-k}}{\binom{N}{n}}, \quad 0 \leq k \leq NF(\eta).$$

Denoting the c.d.f. of hypergeometric distribution as $\Phi_{\text{HG}}(k; N, n, NF(\eta))$, we have

$$\mathbb{P}(\widehat{\eta} \leq \eta) = 1 - \Phi_{\text{HG}}\big(\lceil(n+1)(1-q)\rceil - 1; N, n, NF(\eta)\big).$$

Then, for any $t \in [0, 1]$,

$$\mathbb{P}(\widehat{\text{Cover}} \leq t) = \mathbb{P}\left(\frac{N}{N-n}F(\widehat{\eta}) - \frac{\lceil(1-q)(n+1)\rceil}{N-n} \leq t\right)$$

$$= \mathbb{P}\left(F(\widehat{\eta}) \leq \frac{\lceil(1-q)(n+1)\rceil + (N-n)t}{N}\right).$$

Since $F(\cdot)$ is monotonely increasing,

$$\mathbb{P}(\widehat{\text{Cover}} \leq t) = \mathbb{P}\left(\widehat{\eta} \leq F^{-1}\left(\frac{\lceil(1-q)(n+1)\rceil + (N-n)t}{N}\right)\right),$$

where $F^{-1}(s) = \inf\{\eta\colon F(\eta) \geq s\}$ for any $s \in [0, 1]$. Plugging in the previous results on the distribution of $\widehat{\eta}$, we have

$$\mathbb{P}(\widehat{\text{Cover}} \leq t) = 1 - \Phi_{\text{HG}}\left(\lceil(n+1)(1-q)\rceil - 1; N, n, NF\left(F^{-1}\left(\frac{\lceil(1-q)(n+1)\rceil + (N-n)t}{N}\right)\right)\right)$$

$$= 1 - \Phi_{\text{HG}}\left(\lceil(n+1)(1-q)\rceil - 1; N, n, N\frac{\lceil\lceil(1-q)(n+1)\rceil + (N-n)t\rceil}{N}\right)$$

$$= 1 - \Phi_{\text{HG}}\left(\lceil(n+1)(1-q)\rceil - 1; N, n, \lceil\lceil(1-q)(n+1)\rceil + (N-n)t\rceil\right)$$

$$= 1 - \Phi_{\text{HG}}\left(\lceil(n+1)(1-q)\rceil - 1; N, n, \lceil(1-q)(n+1)\rceil + \lceil(N-n)t\rceil\right)$$

where the second equality uses the fact that $F(\eta) \in \{0, 1/N, \ldots, (N-1)/N, 1\}$, hence $F(F^{-1}(s)) = \lceil Ns\rceil/N$ for any $s \in [0, 1]$. By tower property, such an equation also holds for the unconditional distribution, marginalized over all the randomness. This completes the proof of Theorem 3. $\qquad\square$

## A.2 Additional visualization of test-time coverage

In this part, we provide more visualization of the distributions of test time coverage $\widehat{\text{Cover}}$ under various sample size configurations. We note that such results also apply to standard application of split conformal prediction when the non-conformity score function $V$ is independent of calibration and test samples, so that Assumption 1 is satisfied.

Figures 6 and 7 plot the p.d.f. of $\widehat{\text{Cover}}$ for $\alpha = 0.05$ and $\alpha = 0.1$, respectively, when fixing $n$ and varying the test sample size $m$. The $y$-axis is obtained by computing $\mathbb{P}(t_{k-1} < \widehat{\text{Cover}} \leq t_k)/(t_k - t_{k-1})$ at $x = (t_{k-1} + t_k)/2$ for a sequence of evenly spaced $\{t_k\} \in [0, 1]$. All figures in this paper for p.d.f.s are obtained in the same way. We see that $\widehat{\text{Cover}}$ concentrates more tightly around the target value $1 - \alpha$ as $m$ and $n$ increases.

Figures 8 and 9 plot the p.d.f. of $\widehat{\text{Cover}}$ for $\alpha = 0.05$ and $\alpha = 0.1$, respectively, where we fix $N = m + n$ but vary the calibration sample size $n$. This mimics the situation where the total number of nodes on the graph is fixed, while we may have flexibility in collecting data as the calibration set. We observe a tradeoff between the calibration accuracy determined by $n$ and the test-sample concentration determined by $n$. The distribution of $\widehat{\text{Cover}}$ is more concentrated around $1 - \alpha$ when $m$ and $n$ are relatively balanced.

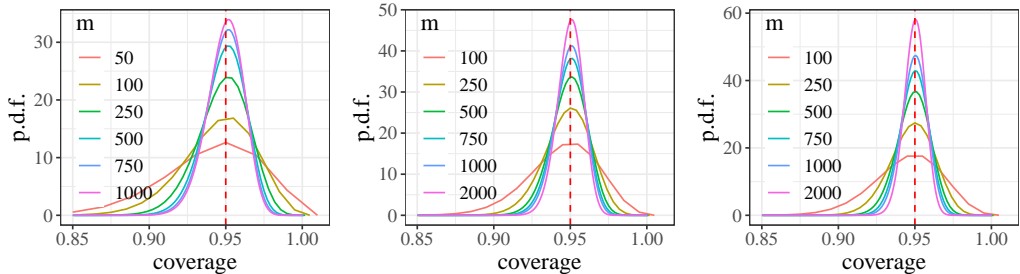

**Figure 6:** P.d.f. of test-time coverage $\widehat{\text{Cover}}$ for $n = 500$ (left), 1000 (middle), 2000 (right) and $\alpha = 0.05$ with curves representing different values of $m$, the test sample size.

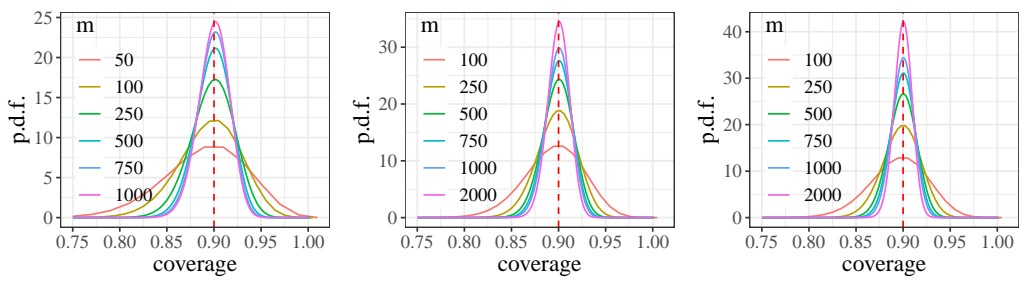

**Figure 7:** P.d.f. of test-time coverage $\widehat{\text{Cover}}$ for $n = 500$ (left), 1000 (middle), 2000 (right) and $\alpha = 0.1$ with curves representing different values of $m$, the test sample size.

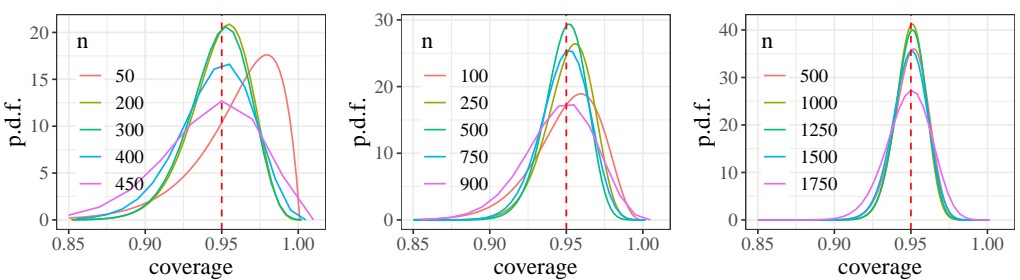

**Figure 8:** P.d.f. of test-time coverage $\widehat{\text{Cover}}$ for $N = m + n = 500$ (left), 1000 (middle), 2000 (right) and $\alpha = 0.05$ with curves representing different values of $n$, the calibration sample size.

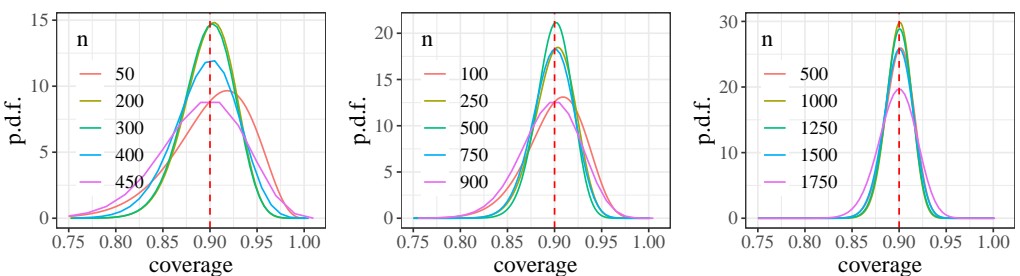

**Figure 9:** P.d.f. of test-time coverage $\widehat{\text{Cover}}$ for $N = m + n = 500$ (left), 1000 (middle), 2000 (right) and $\alpha = 0.1$ with curves representing different values of $n$, the calibration sample size.

# B    Discussion on full conformal prediction, split conformal prediction

In this part, we discuss the relation of our application of conformal prediction to full conformal prediction and split conformal prediction, two prominent conformal prediction methods proposed by Vovk and his coauthors in [43]. Split conformal prediction is mostly widely used due to its computational efficiency, where exchangeability is usually ensured by independence (which is not obvious for graph data) as we discussed briefly in the introduction.

Full conformal prediction (FCP) is arguably the most versatile form of conformal prediction. Given calibration data $Z_i = (X_i, Y_i) \in \mathcal{X} \times \mathcal{Y}$, $i = 1, \ldots, n$, and given a test point $X_{n+1} \in \mathcal{X}$ whose outcome $Y_{n+1} \in \mathcal{Y}$ is unknown, at every hypothesized value $y \in \mathcal{Y}$, FCP uses any algorithm $S$ to train the following scores

$$S_i^y = S(X_i, Y_i; Z_1, \ldots, Z_{i-1}, Z_{i+1}, \ldots, Z_n, (X_{n+1}, y)), \quad i = 1, \ldots, n,$$

where $S$ is symmetric in the arguments $Z_1, \ldots, Z_{i-1}, Z_{i+1}, \ldots, Z_n, (X_{n+1}, y)$, as well as

$$S_{n+1}^y = S(X_{n+1}, y; Z_1, \ldots, Z_n).$$

Here, for $1 \le i \le n$, $S_i^y$ intuitively measures how well the observation $(X_i, Y_i)$ conforms to the observations $Z_1, \ldots, Z_{i-1}, Z_{i+1}, \ldots, Z_n, (X_{n+1}, y)$ with the hypothesized value of $y$. For instance, when using a linear prediction model, it can be chosen as the prediction residual

$$S_i^y = |Y_i - X_i^\top \widehat{\theta}^y|,$$

where $\widehat{\theta}^y$ is the ordinary least squares coefficient by a linear regression of $Y_1, \ldots, Y_{i-1}, Y_{i+1}, \ldots, Y_n, y$ over $X_1, \ldots, X_{i-1}, X_{i+1}, \ldots, X_n, X_{n+1}$. More generally, one may train a prediction model $\widehat{\mu}^y \colon \mathcal{X} \to \mathcal{Y}$ using $Z_1, \ldots, Z_{i-1}, Z_{i+1}, \ldots, Z_n, (X_{n+1}, y)$, and set $S_i^y = |Y_i - \widehat{\mu}^y(X_i)|$. For a confidence level $\alpha \in (0, 1)$, the FCP prediction set is then

$$\widehat{C}(X_{n+1}) := \left\{ y \colon \frac{1 + \mathbb{1}\{S_i^y > S_{n+1}^y\}}{n+1} \le \alpha \right\}.$$

Since the original form of FCP involves training $n + 1$ models at each hypothesized value $y$, its computation can be very intense. It is thus impractical to directly apply FCP to GNN models (i.e., imagining $S$ as the GNN training process on the entire graph with a hypothesized outcome $y$).

Split conformal prediction (SCP) is a computationally-efficient special case of FCP that is most widely used for i.i.d. data. The idea is to set aside an independent fold of data to output a single trained model. To be specific, we assume access to a given non-conformity score $V \colon \mathcal{X} \times \mathcal{Y} \to \mathbb{R}$, i.i.d. calibration data $Z_i = (X_i, Y_i)_{i=1}^n$, and an independent test sample $(X_{n+1}, Y_{n+1})$ from the same distribution with $Y_{n+1}$ unobserved. Here by a "given" score, we mean that it is obtained without knowing the calibration and test sample; usually, it is trained on an independent set of data $\{(X_j, Y_j)\}_{j \in \mathcal{D}_{\text{train}}}$ before seeing the calibration and test sample. Then define $V_i = V(X_i, Y_i)$ for $i = 1, \ldots, n$. The SCP prediction set is

$$\widehat{C}(X_{n+1}) = \left\{ y \colon \frac{1 + \mathbb{1}\{V_i > V(X_{n+1}, y)\}}{n+1} \le \alpha \right\}.$$

The above set is usually convenient to compute, because we only need one single model to obtain $V$. The validity of SCP usually relies on the independence of $V$ to calibration and test data as we mentioned in the introduction. However, the application of SCP to GNN model is also not straightforward: as we discussed in the main text, the model training step already uses the calibration and test samples, and the nodes are correlated.

Indeed, our method can be seen as a middle ground between FCP and SCP: it only requires one single prediction model as SCP does, but allows to use calibration and test data in the training step as FCP does. In our method introduced in the main text, there exists a fixed function $V \colon \mathcal{Y} \times \mathcal{Y} \to \mathbb{R}$ (provided by APS and CQR) such that

$$S_i^y = V(\widehat{\mu}(X_i), Y_i), \quad S_{n+1}^y = V(\widehat{\mu}(X_{n+1}), y),$$

where $\widehat{\mu}$ is the final output from the second GNN model whose training process does not utilize the outcomes $Y_1, \ldots, Y_n$ and $y$, but uses the features $X_1, \ldots, X_n$ and $X_{n+1}$.

**Algorithm 1:** Pseudo-code for CF-GNN algorithm.

---

**Input**: Graph $G = (\mathcal{V}, \mathcal{E}, \mathbf{X})$; a trained base GNN model $\text{GNN}_\theta$; non-conformity score function $V(X, Y)$; pre-specified mis-coverage rate $\alpha$, Randomly initialized $\vartheta$ for the conformal correction model $\text{GNN}_\vartheta$.

**while** *not done* **do**

    **for** *i in {1, ..., $|\mathcal{V}_{\text{cor-calib}} \cup \mathcal{V}_{\text{cor-test}}|$}* **do**

        $\widehat{\mu}(X_i) = \text{GNN}_\theta(\mathbf{X_i}, G)$               `// Base GNN output scores`

        $\tilde{\mu}(X_i) = \text{GNN}_\vartheta(\widehat{\mu}(X_i), G)$      `// Correction model output scores`

    **end**

    $n, m = |\mathcal{V}_{\text{cor-calib}}|, |\mathcal{V}_{\text{cor-test}}|$        `// Size of correction calib/test set`

    $\widehat{\alpha} = \frac{1}{n+1} * \alpha$                   `// Finite-sample correction`

    $\widehat{\eta} = \text{DiffQuantile}(\{V(X_i, Y_i) | i \in \mathcal{V}_{\text{cor-calib}}\})$ `// Compute non-conformity scores`

    **if** Classification **then**

        $\mathcal{L}_{\text{Ineff}} = \frac{1}{m} \sum_{i \in \mathcal{V}_{\text{cor-test}}} \frac{1}{|\mathcal{Y}|} \sum_{k \in \mathcal{Y}} \sigma(\frac{V(X_i, k) - \widehat{\eta}}{\tau})$

                               `// Inefficiency proxy for classification tasks`

    **end**

    **if** Regression **then**

        $\mathcal{L}_{\text{Ineff}} = \frac{1}{m} \sum_{i \in \mathcal{V}_{\text{cor-test}}} (\tilde{\mu}_{1-\alpha/2}(X)_i + \widehat{\eta}) - (\tilde{\mu}_{\alpha/2}(X)_i - \widehat{\eta})$

                               `// Inefficiency proxy for regression tasks`

        $\mathcal{L}_{\text{Ineff}} += \gamma \frac{1}{m} \sum_{i \in \mathcal{V}_{\text{cor-test}}} (\tilde{\mu}_{1-\alpha/2}(X)_i - \widehat{\mu}_{1-\alpha/2}(X)_i)^2 + (\tilde{\mu}_{\alpha/2}(X)_i - \widehat{\mu}_{\alpha/2}(X)_i)^2$

                               `// Consistency regularization term`

    **end**

    $\vartheta \leftarrow \vartheta - \nabla_\vartheta \mathcal{L}_{\text{Ineff}}$           `// Optimizing` $\vartheta$ `to reduce inefficiency`

**end**

---

## C   Algorithm overview

We describe the pseudo-code of CF-GNN in Algorithm 1.

## D   Deferred details for experiments

### D.1   Hyperparameters

Table 5 reports our set of hyperparameter ranges. We conduct 100 iterations of Bayesian Optimization for CF-GNN with the validation set inefficiency proxy as the optimization metric. To avoid overfitting, each iteration only uses the first GNN run. The optimized hyperparameters are then used for all 10 GNN runs and we then reported the average and standard deviation across runs. Each experiment is done with a single NVIDIA 2080 Ti RTX 11GB GPU.

**Table 5:** Hyperparameter range for CF-GNN.

| Task | Param. | Range |
|---|---|---|
| Classification | $\text{GNN}_\vartheta$ Hidden dimension | [16,32,64,128,256] |
| | Learning rate | [1e-1, 1e-2, 1e-3, 1e-4] |
| | $\text{GNN}_\vartheta$ Number of GNN Layers | [1,2,3,4] |
| | $\text{GNN}_\vartheta$ Base Model | [GCN, GAT, GraphSAGE, SGC] |
| | $\tau$ | [10, 1, 1e-1, 1e-2, 1e-3] |
| Regression | $\text{GNN}_\vartheta$ Hidden dimension | [16,32,64,128,256] |
| | Learning rate | [1e-1, 1e-2, 1e-3, 1e-4] |
| | $\text{GNN}_\vartheta$ Number of GNN Layers | [1, 2, 3, 4] |
| | $\text{GNN}_\vartheta$ Base Model | [GCN, GAT, GraphSAGE, SGC] |
| | Reg. loss coeff. $\gamma$ | [1, 1e-1] |

## D.2 Baseline Details

We report the details about baselines below and the hyperparameter range in Table 6.

1. Temperature Scaling [13] divides the logits with a learnable scalar. It is optimized over NLL loss in the validation set.

2. Vector Scaling [13] has a scalar to scale the logits for each class dimension and adds an additional classwide bias. It is optimized over NLL loss in the validation set.

3. Ensemble Temperature Scaling [49] learns an ensemble of uncalibrated, temperature-scaled calibrated calibrators.

4. CaGCN [44] uses an additional GCN model that learns a temperature scalar for each node based on its neighborhood information.

5. GATS [17] identifies five factors that affect GNN calibration and designs a model that accounts for these five factors by using per-node temperature scaling and attentive aggregation from the local neighborhood.

6. QR [25] uses a pinball loss to produce quantile scores. It is CQR without the conformal prediction adjustment.

7. MC dropout [9] turns on dropout during evaluation and produces $K$ predictions. We then take the 95% quantile of the predicted distribution. We also experimented with taking a 95% confidence interval but 95% quantile has better coverage, thus we adopt the quantile approach.

8. BayesianNN [23] model the label with normal distribution and the model produces two heads, where one corresponds to the mean and the second log variance. We then calculate the standard deviation as the square root of the exponent of log variance. Then we take the [mean-1.96*standard deviation, mean+1.96*standard deviation] for the 95% interval.

**Table 6:** Hyperparameter range for baselines.

| Baseline | Param. | Range |
|---|---|---|
| Temperature Scaling | No hyperparameter | Not Applicable |
| Vector Scaling | No hyperparameter | Not Applicable |
| Ensemble Temp Scaling | No hyperparameter | Not Applicable |
| CaGCN | Dropout
Hidden dimension
Number of GNN Layers
Weight Decay | [0.3, 0.5, 0.7]
[16, 32, 64, 128, 256]
[1,2,3,4]
[0, 1e-3, 1e-2, 1e-1] |
| GATS | Dropout
Hidden dimension
Number of GNN Layers
Weight Decay | [0.3, 0.5, 0.7]
[16, 32, 64, 128, 256]
[1,2,3,4]
[0, 1e-3, 1e-2, 1e-1] |
| MC Dropout | Number of Predictions | [100, 500, 1,000] |
| BayesianNN | No hyperparameter | Not Applicable |

## D.3 Dataset

For node classification, we use the common node classification datasets in Pytorch Geometric package. For node regression, we use datasets in [20]. We report the dataset statistics at Table 7.

## D.4 Marginal coverage and inefficiency across GNN architectures

We additionally conduct marginal coverage and inefficiency comparisons of CF-GNN over the vanilla CP across 4 different GNN architectures: GCN, GAT, GraphSAGE, and SGC. The result for marginal

**Table 7:** Dataset statistics.

| Domain | Dataset | Task | # Nodes | # Edges | # Features | # Labels |
|---|---|---|---|---|---|---|
| Citation | Cora | Classification | 2,995 | 16,346 | 2,879 | 7 |
| | DBLP | Classification | 17,716 | 105,734 | 1,639 | 4 |
| | CiteSeer | Classification | 4,230 | 10,674 | 602 | 6 |
| | PubMed | Classification | 19,717 | 88,648 | 500 | 3 |
| Co-purchase | Computers | Classification | 13,752 | 491,722 | 767 | 10 |
| | Photos | Classification | 7,650 | 238,162 | 745 | 8 |
| Co-author | CS | Classification | 18,333 | 163,788 | 6,805 | 15 |
| | Physics | Classification | 34,493 | 495,924 | 8,415 | 5 |
| Transportation | Anaheim | Regression | 914 | 3,881 | 4 | – |
| | Chicago | Regression | 2,176 | 15,104 | 4 | – |
| Geography | Education | Regression | 3,234 | 12,717 | 6 | – |
| | Election | Regression | 3,234 | 12,717 | 6 | – |
| | Income | Regression | 3,234 | 12,717 | 6 | – |
| | Unemployment | Regression | 3,234 | 12,717 | 6 | – |
| Social | Twitch | Regression | 1,912 | 31,299 | 3,170 | – |

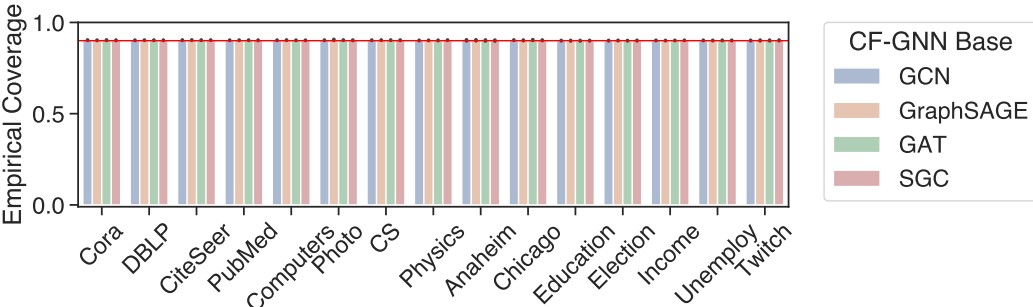

**Figure 10:** Empirical coverage across 15 datasets with 10 independent runs of GNN, using CF-GNN.

coverage is in Figure 10. The result for inefficiency is in Table 8. We observe consistent improvement in inefficiency reduction across these architectures, suggesting CF-GNN is a GNN-agnostic efficiency improvement approach.

### D.5 CF-GNN with Regularized Adaptive Prediction Sets

To further showcase that CF-GNN is a versatile framework that adapts to any advancement in non-conformity scores, we experiment on RAPS [2], which regularizes APS to produce a smaller prediction set size. We report the performance using the GCN backbond in Table 9. We observe that CF-GNN still obtains impressive inefficiency reduction compared to the vanilla application of RAPS to GNN.

### D.6 Conditional coverage on full set of network features

We report the full set of network features and calculate the worse-slice coverage in Table 10 for target coverage of 0.9 and Table 11 for a target coverage of 0.95. We observe that CF-GNN achieves satisfactory conditional coverage across a wide range of diverse network features.

### D.7 Prediction accuracy versus uncertainty calibration

As we discussed in the main text, the original GNN trained towards optimal prediction accuracy does not necessarily yield the most efficient prediction model; this is corrected by the second GNN in CF-GNN which improves the efficiency of conformal prediction sets/intervals. With our approach, one can use the output of the original GNN for point prediction while that of the second GNN for efficient uncertainty quantification, without necessarily overwriting the first accurate prediction model. However, a natural question here still remains, which is that whether applying the second

**Table 8:** Empirical inefficiency measured by the size/length of the prediction set/interval for node classification/regression. The result uses APS for classification and CQR for regression. We report the average and standard deviation calculated from 10 GNN runs with each run of 100 conformal splits.

| Task | GNN Model | GCN | GraphSAGE | GAT | SGC |
|------|-----------|-----|-----------|-----|-----|
| | Dataset | CP——→CF-GNN | CP——→CF-GNN | CP——→CF-GNN | CP——→CF-GNN |
| Node classif. | Cora | $3.80_{\pm.28}\xrightarrow{-58.44\%}1.58_{\pm.22}$ | $6.73_{\pm.19}\xrightarrow{-76.50\%}1.58_{\pm.15}$ | $4.14_{\pm.16}\xrightarrow{-62.53\%}1.55_{\pm.10}$ | $3.88_{\pm.19}\xrightarrow{-62.23\%}1.47_{\pm.10}$ |
| | DBLP | $2.43_{\pm.03}\xrightarrow{-49.20\%}1.23_{\pm.01}$ | $3.91_{\pm.01}\xrightarrow{-68.27\%}1.24_{\pm.01}$ | $2.02_{\pm.06}\xrightarrow{-37.51\%}1.26_{\pm.01}$ | $2.44_{\pm.04}\xrightarrow{-48.93\%}1.24_{\pm.02}$ |
| | CiteSeer | $3.86_{\pm.11}\xrightarrow{-74.50\%}0.99_{\pm.01}$ | $5.88_{\pm.02}\xrightarrow{-83.07\%}1.00_{\pm.01}$ | $3.18_{\pm.25}\xrightarrow{-68.56\%}1.00_{\pm.01}$ | $3.79_{\pm.14}\xrightarrow{-73.43\%}1.01_{\pm.02}$ |
| | PubMed | $1.60_{\pm.02}\xrightarrow{-19.44\%}1.29_{\pm.04}$ | $1.93_{\pm.28}\xrightarrow{-36.95\%}1.22_{\pm.03}$ | $1.37_{\pm.02}\xrightarrow{-10.54\%}1.23_{\pm.02}$ | $1.60_{\pm.02}\xrightarrow{-24.27\%}1.21_{\pm.02}$ |
| | Computers | $3.56_{\pm.13}\xrightarrow{-50.22\%}1.77_{\pm.11}$ | $6.00_{\pm.10}\xrightarrow{-45.74\%}3.26_{\pm.48}$ | $2.33_{\pm.11}\xrightarrow{-8.18\%}2.14_{\pm.12}$ | $3.44_{\pm.14}\xrightarrow{-45.69\%}1.87_{\pm.15}$ |
| | Photo | $3.79_{\pm.13}\xrightarrow{-57.03\%}1.63_{\pm.17}$ | $4.52_{\pm.47}\xrightarrow{-37.22\%}2.84_{\pm.67}$ | $2.24_{\pm.21}\xrightarrow{-19.38\%}1.81_{\pm.16}$ | $3.81_{\pm.14}\xrightarrow{-62.86\%}1.41_{\pm.05}$ |
| | CS | $7.79_{\pm.29}\xrightarrow{-55.83\%}3.44_{\pm.33}$ | $14.68_{\pm.02}\xrightarrow{-88.63\%}1.67_{\pm.14}$ | $6.87_{\pm.48}\xrightarrow{-73.08\%}1.85_{\pm.10}$ | $7.76_{\pm.25}\xrightarrow{-73.92\%}2.02_{\pm.22}$ |
| | Physics | $3.11_{\pm.07}\xrightarrow{-65.36\%}1.08_{\pm.10}$ | $4.91_{\pm.01}\xrightarrow{-72.97\%}1.33_{\pm.08}$ | $2.00_{\pm.19}\xrightarrow{-45.23\%}1.09_{\pm.06}$ | $3.10_{\pm.08}\xrightarrow{-57.22\%}1.32_{\pm.12}$ |
| Average Improvement | | -53.75% | -63.75% | -40.63% | -56.07% |
| Node regress. | Anaheim | $2.89_{\pm.39}\xrightarrow{-25.00\%}2.17_{\pm.11}$ | $2.37_{\pm.05}\xrightarrow{-23.12\%}1.82_{\pm.07}$ | $3.12_{\pm.38}\xrightarrow{-31.27\%}2.14_{\pm.11}$ | $2.94_{\pm.24}\xrightarrow{-24.90\%}2.21_{\pm.16}$ |
| | Chicago | $2.05_{\pm.07}\xrightarrow{-0.48\%}2.04_{\pm.17}$ | $2.08_{\pm.05}\xrightarrow{-7.90\%}1.92_{\pm.09}$ | $1.95_{\pm.04}\xrightarrow{-68.15\%}0.62_{\pm.93}$ | $2.02_{\pm.03}\xrightarrow{-1.37\%}1.99_{\pm.07}$ |
| | Education | $2.56_{\pm.02}\xrightarrow{-5.07\%}2.43_{\pm.05}$ | $2.20_{\pm.04}\xrightarrow{+8.44\%}2.38_{\pm.08}$ | $2.48_{\pm.05}\xrightarrow{-2.76\%}2.41_{\pm.04}$ | $2.55_{\pm.02}\xrightarrow{-2.80\%}2.48_{\pm.04}$ |
| | Election | $0.90_{\pm.01}\xrightarrow{+0.21\%}0.90_{\pm.02}$ | $0.87_{\pm.01}\xrightarrow{-0.80\%}0.86_{\pm.02}$ | $0.89_{\pm.00}\xrightarrow{-1.23\%}0.88_{\pm.02}$ | $0.90_{\pm.00}\xrightarrow{-0.42\%}0.90_{\pm.02}$ |
| | Income | $2.51_{\pm.12}\xrightarrow{-4.58\%}2.40_{\pm.05}$ | $2.08_{\pm.04}\xrightarrow{+32.23\%}2.75_{\pm.23}$ | $2.35_{\pm.03}\xrightarrow{-0.23\%}2.34_{\pm.07}$ | $2.42_{\pm.01}\xrightarrow{+3.04\%}2.49_{\pm.04}$ |
| | Unemploy. | $2.72_{\pm.03}\xrightarrow{-10.83\%}2.43_{\pm.04}$ | $2.75_{\pm.06}\xrightarrow{-12.90\%}2.39_{\pm.05}$ | $2.80_{\pm.08}\xrightarrow{-14.56\%}2.40_{\pm.04}$ | $2.72_{\pm.02}\xrightarrow{-11.05\%}2.42_{\pm.04}$ |
| | Twitch | $2.43_{\pm.10}\xrightarrow{-1.36\%}2.39_{\pm.07}$ | $2.48_{\pm.09}\xrightarrow{-3.06\%}2.40_{\pm.07}$ | $2.50_{\pm.14}\xrightarrow{-5.53\%}2.36_{\pm.07}$ | $2.42_{\pm.08}\xrightarrow{-1.43\%}2.38_{\pm.06}$ |
| Average Improvement | | -6.73% | -1.02% | -17.68% | -5.56% |

**Table 9:** Comparison with other non-conformity scores that reduce inefficiency.

| Size | CP ⟶ CF-GNN |
|------|-------------|
| Cora | $1.67_{\pm.11}\xrightarrow{-15.35\%}1.42_{\pm.05}$ |
| DBLP | $1.39_{\pm.02}\xrightarrow{-5.00\%}1.32_{\pm.01}$ |
| CiteSeer | $1.30_{\pm.07}\xrightarrow{-19.85\%}1.04_{\pm.04}$ |
| PubMed | $1.23_{\pm.01}\xrightarrow{+2.40\%}1.26_{\pm.02}$ |
| Computers | $1.58_{\pm.02}\xrightarrow{-4.59\%}1.51_{\pm.05}$ |
| Photo | $1.34_{\pm.01}\xrightarrow{-10.47\%}1.20_{\pm.01}$ |
| CS | $1.29_{\pm.04}\xrightarrow{-6.13\%}1.21_{\pm.02}$ |

GNN drastically changes the prediction accuracy. This question is more relevant to the classification problem since for regression our method only adjusts the confidence band. For classification, we consider top-1 class prediction as the "point prediction". We present its accuracy "Before" and "After" the correction in Table 12, which shows that this correction typically does not result in a visible change in accuracy. In addition, in a new experiment on Cora, we find that 100% of the top-1 class from the base GNN are in CF-GNN's prediction sets. The potential to develop steps that explicitly consider point prediction accuracy is an exciting avenue for future research.

# E  Extended Related Works

**Uncertainty quantification for graph neural networks.** Uncertainty quantification (UQ) is a well-studied subject in general machine learning and also recently in GNNs. For multi-class classification, the raw prediction scores are often under/over-confident and thus various calibration methods are proposed for valid uncertainty estimation such as temperate scaling [13], vector scaling [13], ensemble temperate scaling [49], and so on [14, 27, 39, 1]. Recently, specialized calibration methods that leverage network principles such as homophily have been developed: examples include CaGCN [44] and GATS [17]. In regression, various methods have been proposed to construct prediction intervals, such as quantile regression [25, 41, 38], bootstrapping with subsampling, model ensembles, and dropout initialization [9, 28, 26, 35], and bayesian approaches with strong modeling assumptions on parameter and data distributions [23, 19]. However, these UQ methods can fail to provide statistically

**Table 10:** CF-GNN achieves conditional coverage. We use Cora/Twitch as an example classification/regression dataset.

| Target: 0.9 | Classification | | Regression | |
|---|---|---|---|---|
| Model | CP | CF-GNN | CP | CF-GNN |
| Marginal Cov. | $0.90_{\pm.02}$ | $0.90_{\pm.01}$ | $0.91_{\pm.02}$ | $0.91_{\pm.03}$ |
| Cond. Cov. (Input Feat.) | $0.89_{\pm.04}$ | $0.90_{\pm.03}$ | $0.90_{\pm.07}$ | $0.86_{\pm.08}$ |
| Cond. Cov. (Cluster) | $0.82_{\pm.07}$ | $0.89_{\pm.03}$ | $0.90_{\pm.06}$ | $0.88_{\pm.07}$ |
| Cond. Cov. (Between) | $0.82_{\pm.06}$ | $0.89_{\pm.03}$ | $0.86_{\pm.08}$ | $0.88_{\pm.07}$ |
| Cond. Cov. (PageRank) | $0.71_{\pm.08}$ | $0.87_{\pm.05}$ | $0.87_{\pm.09}$ | $0.89_{\pm.07}$ |
| Cond. Cov. (Load) | $0.83_{\pm.05}$ | $0.90_{\pm.03}$ | $0.86_{\pm.08}$ | $0.88_{\pm.07}$ |
| Cond. Cov. (Harmonic) | $0.89_{\pm.04}$ | $0.87_{\pm.05}$ | $0.88_{\pm.08}$ | $0.91_{\pm.06}$ |
| Cond. Cov. (Degree) | $0.79_{\pm.05}$ | $0.89_{\pm.04}$ | $0.86_{\pm.08}$ | $0.89_{\pm.06}$ |

**Table 11:** CF-GNN achieves conditional coverage. We use Cora/Twitch as an example classification/regression dataset.

| Target: 0.95 | Classification | | Regression | |
|---|---|---|---|---|
| Model | CP | CF-GNN | CP | CF-GNN |
| Marginal Cov. | $0.95_{\pm.01}$ | $0.95_{\pm.01}$ | $0.96_{\pm.02}$ | $0.96_{\pm.02}$ |
| Cond. Cov. (Input Feat.) | $0.94_{\pm.02}$ | $0.94_{\pm.03}$ | $0.95_{\pm.04}$ | $0.94_{\pm.05}$ |
| Cond. Cov. (Cluster) | $0.89_{\pm.06}$ | $0.93_{\pm.04}$ | $0.96_{\pm.03}$ | $0.96_{\pm.03}$ |
| Cond. Cov. (Between) | $0.81_{\pm.06}$ | $0.95_{\pm.03}$ | $0.94_{\pm.05}$ | $0.94_{\pm.05}$ |
| Cond. Cov. (PageRank) | $0.78_{\pm.06}$ | $0.94_{\pm.03}$ | $0.94_{\pm.05}$ | $0.94_{\pm.05}$ |
| Cond. Cov. (Load) | $0.81_{\pm.06}$ | $0.94_{\pm.03}$ | $0.94_{\pm.05}$ | $0.95_{\pm.05}$ |
| Cond. Cov. (Harmonic) | $0.88_{\pm.04}$ | $0.95_{\pm.03}$ | $0.96_{\pm.04}$ | $0.95_{\pm.04}$ |
| Cond. Cov. (Degree) | $0.83_{\pm.05}$ | $0.88_{\pm.06}$ | $0.94_{\pm.04}$ | $0.94_{\pm.04}$ |

rigorous and empirically valid coverage guarantee (see Table 1). In contrast, CF-GNN achieves valid marginal coverage in both theory and practice. Uncertainty quantification has also been leveraged to deal with out-of-distribution detection and imbalanced data in graph neural networks [50, 10]. While it is not the focus here, we remark that conformal prediction can also be extended to tackle such issues [18], and it would be interesting to explore such applications for graph data.

**Conformal prediction for graph neural networks.** As we discussed, the application of conformal prediction to graph-structured data remains largely unexplored. At the time of submission, the only work we awared of is [7], who claims that nodes in the graph are not exchangeable in the inductive setting and employs the framework of [3] to construct conformal prediction sets using neighborhood nodes as the calibration data. In contrast, we study the transductive setting where certain exchangeablility property holds and allows for flexibility in the training step. We also study the efficiency aspect that is absent in [7]. In addition, there have been concurrent works [15, 33] that observe similar exchangeability and validity of conformal prediction in either transductive setting or other network models. In particular, [15] proposes a diffusion-based method that aggregates non-conformity scores of neighbor nodes to improve efficiency, while our approach learns the aggregation of neighbor scores, which is more general than their approach. [32] studies the exchangeability for node regression under certain network models instead of our transductive setting with GNNs, and without considering the efficiency aspect. With a growing recent interest in conformal prediction for graphs, there are even more recent works that focus on validity [32] and link prediction [34].

**Efficiency of conformal prediction.** While conformal prediction enjoys distribution-free coverage for any non-conformity score based on any prediction model, its efficiency (i.e., size of prediction sets or length of prediction intervals) varies with specific choice of the scores and models. How to achieve desirable properties such as efficiency is a topic under intense research in conformal prediction. To this end, one major thread designs good non-conformity scores such as APS [37] and CQR [36]. More recent works take another approach, by modifying the training process of the prediction model to

**Table 12:** CF-GNN does not change the top-1 class prediction accuracy for classification tasks.

| Dataset | Before | After |
|---------|--------|-------|
| Cora | $0.844_{\pm 0.004}$ | $0.843_{\pm 0.016}$ |
| DBLP | $0.835_{\pm 0.001}$ | $0.832_{\pm 0.002}$ |
| CiteSeer | $0.913_{\pm 0.002}$ | $0.911_{\pm 0.002}$ |

further improve efficiency. This work falls into the latter case. Our idea applies to any non-conformity scores, as demonstrated with APS and CQR, two prominent examples of the former case. Related to our work, ConfTr [40] also simulates conformal prediction so as to train a prediction model that eventually leads to more efficient conformal prediction sets. However, our approach differs from theirs in significant ways. First, ConfTr modifies model training, while CF-GNN conducts post-hoc correction without changing the original prediction. Second, ConfTr uses the training set to simultaneously optimize model prediction and efficiency of conformal prediction, while we withhold a fraction of calibration data to optimize the efficiency. Third, our approach specifically leverages the rich topological information in graph-structured data to achieve more improvement in efficiency. Finally, we also propose a novel loss for efficiency in regression tasks.

