# OpenReview forum: "Uncertainty Quantification over Graph with Conformalized Graph Neural Networks"
_NeurIPS.cc/2023/Conference — NeurIPS 2023 spotlight_

### Official Review · Reviewer_rkWb · 2023-06-30

**Soundness:** 3 good
**Presentation:** 3 good
**Contribution:** 3 good
**Rating:** 7
**Confidence:** 3

**Summary:**

Conformal Prediction (CP) outputs a prediction set that contains the true label with a certain likelihood given assumptions on exchangeability. It is a well-known and popular uncertainty quantification (UQ) technique.  The authors propose a technique that unites GNNs and CP called conformalized GNN (CF-GNN). The technique is crucially permutation equivariant and topology aware.

**Strengths:**

The paper tackles an interesting problem of unifying CP and GNNs. UQ is an important area of research and CP offers strong sound guarantees to tackle such a problem. The paper is well-written and I appreciated that the authors keep important considerations such as permutation equivariance, efficiency and topology in mind when developing their technique. The evaluation is extensive and shows good results.

**Weaknesses:**

The technique focuses on a transductive setting only. While this is still a valid and interesting setting, the fact that the technique focuses on this setting only is a weakness.

**Questions:**

Is there a reason for the focus on the transductive setting? Are there any clear challenges in inductive node-level or inductive graph-level tasks for instance. I believe inductive graph-level tasks to be an extremely interesting approach as doing UQ on molecular predictions may be a very important application. Another important application could also be link prediction.

**Limitations:**

The authors address the transductive focus as a limitation which is an important consideration. I don't foresee potential negative societal impact.

---

> ### Author Rebuttal · Authors · 2023-08-09
>
> We thank the reviewer for the insightful feedback and for acknowledging that our paper tackles an interesting problem, that it is well-written, and that the evaluation is extensive. The reviewer raises great questions and we respond to them below:
>
> > Why focus only on transductive settings? Extension to inductive and graph-level tasks?
>
> We thank the reviewer for this intriguing comment. The transductive setting is a widely useful evaluation setup for graph machine learning [1,2]. In Section 3, we show in theory that graph exchangeability holds for transductive settings, which enable the direct application of conformal prediction to graph-structured data. While other settings such as inductive learning are at least equally exciting, we start with the transductive setting to connect with the most standard conformal inference approach and establish the foundation for conformal prediction on graphs.
>
> There are indeed nontrivial challenges for the inductive node-level setting: it may need a completely different setup, theoretical analysis, and methodology, which goes way beyond the scope of this paper. In the inductive setting, newly arrived testing nodes will be potentially connected to calibration nodes in the graph. Thus the non-conformity scores of calibration nodes are dependent on the ordering of testing nodes. In other words, different ordering of testing nodes that connect to the graph may imply that calibration nodes have different non-conformity scores. This violates the permutation invariance condition we laid out for graph exchangeability. We believe that the extension to the inductive setting needs significantly different techniques (the reviewer has pointed out that this is a very valuable future direction to pursue). A possibility is to extend the “beyond exchangeability” framework [3] by reweighting conformity scores by using certain predefined “similarity scores”. However, even the theory in [3] might not apply since it needs data-independent weights as input, and this precludes the use of network structures.
>
> For the graph-level prediction task such as the prediction of molecule properties, each data point is a graph. Thus, there is no issue with graph exchangeability since there are no direct dependencies between calibration and testing graphs, as seen in the node-level prediction problem. In this problem, conventional conformal prediction methods can be directly applied.
>
> We will expand the discussion on these points in the updated paper.
>
> [1] Kipf, T. N., & Welling, M. (2016). Semi-supervised classification with graph convolutional networks. ICLR.
>
> [2] Hamilton, W., Ying, Z., & Leskovec, J. (2017). Inductive representation learning on large graphs. NeurIPS.
>
> [3] Barber, R. F., Candes, E. J., Ramdas, A., & Tibshirani, R. J. (2023). Conformal prediction beyond exchangeability. The Annals of Statistics, 51(2), 816-845.

---

> > ### Comment · Reviewer_rkWb · 2023-08-11
> >
> > Thank you for your interesting response and for adding a discussion regarding this point in the paper. I think this is great work and addresses a very valuable problem. I have raised my score accordingly.

---

### Official Review · Reviewer_y5h3 · 2023-07-04

**Soundness:** 3 good
**Presentation:** 3 good
**Contribution:** 3 good
**Rating:** 7
**Confidence:** 2

**Summary:**

The authors propose Conformalized Graph Neural Networks (CF-GNNs), which extends conformal prediction to graphs for uncertainty quantification. The framework allows a GNN to produce confidence intervals for its predictions, based on an uncertainty estimation on a withhold calibration set. Under permutation invariance condition, they provide theoretical guarantee for the coverage of test-time confidence intervals. Furthermore, a learnable correction model is introduced to empirically produce more efficient prediction intervals. Experiments show that CF-GNNs acheive a better coverage rate over baselines and meanwhile have a more efficient (smaller) confidence interval than a naive conformal prediction.

**Strengths:**

- [originality] the authors extend conformal prediction to node-level task and give theoretical coverage guarantee. As far as I know, this adaption and theorical results are new.
- [significance] this work makes solid theoretical and empirical contributions to uncertainty quantifications of graphs. The proposed method has a greater coverage rate and a smaller prediction intervals size.

**Weaknesses:**

See Questions.

**Questions:**

- At Line 276 the authors mention "As smaller coverage always leads to higher efficiency, for a fair comparison, we can only compare methods on efficiency that achieve the same coverage. Thus, we do not evaluate UQ baselines here since they do not produce exact coverage and are thus not comparable". Do you think there is a way to make some of the other UQ methods empirically reach $(1-\alpha)$ coverage so that we can get some senses of their size of prediction sets for comparison?

**Limitations:**

No limitation discussed. No potential negative social impact.

---

> ### Author Rebuttal · Authors · 2023-08-09
>
> We thank the reviewer for insightful feedback and for acknowledging that our paper is original and makes solid theoretical and empirical contributions. The reviewer has raised great questions and we respond to them below:
>
> > Ways to make baseline UQ methods empirically reach $1-\alpha$ coverage?
>
> We thank the reviewers for raising this great question! We want to emphasize that, first, these UQ methods cannot reach nominal coverage either in theory or empirically. In contrast, what is unique about CF-GNN is that it rigorously achieves any pre-specified coverage rate. Second, even if it is possible to tune the nominal level of previous UQ methods to achieve $1-\alpha$ coverage and compare their efficiency with CF-GNN, in practice, it is impossible to know to which extent one should adjust the nominal level for those methods. On a side note, conformal methods could be used to calibrate the nominal level for these UQ methods to achieve valid coverage; however, it is beyond the original implementation of the UQ method and requires substantial effort in method development.
>
> However, for the sake of evaluating the methods with access to gold labels, it is possible to reach $(1-\alpha)$ coverage by tuning the nominal coverage level of the previous methods and selecting the level when the empirical coverage reaches the target coverage. For our preliminary experiment, we used the Cora dataset with a target coverage of 95% and we studied Graph Attention Temperature Scaling (GATS) [NeurIPS 2022], which represents the most recent GNN-based UQ calibration method in our set of baselines. We conducted a grid search of nominal coverage from 90% to 100% with step size 0.1% and stopped when it reached the empirical target coverage. The stopping nominal coverage is adjusted to $1-\alpha$ = 92.2%, which is far away from the pre-specified target 95% coverage. The prediction set size becomes 1.82 which is still larger (less efficient) than CF-GNN, which has size 1.76. We will discuss this interesting point in the revised manuscript.

---

### Official Review · Reviewer_XvAj · 2023-07-07

**Soundness:** 3 good
**Presentation:** 4 excellent
**Contribution:** 3 good
**Rating:** 7
**Confidence:** 3

**Summary:**

This paper presents a new approach, known as conformalized Graph Neural Networks (CF-GNN), designed to bring reliable uncertainty estimates to graph-structured data prediction models. The study's primary contribution is the innovative adaptation of conformal prediction (CP) to Graph Neural Networks (GNNs). The proposed CF-GNN model is capable of generating prediction sets or intervals that encapsulate the true label, with a level of coverage probability (90%) that can be predefined.

**Strengths:**

The idea of extending conformal prediction to GNNs is quite interesting and timely. The authors establish a permutation invariance condition that justifies the application of CP on graph data and gives a precise outline of test-time coverage. The empirical evidence from numerous experiments validates the effectiveness of CF-GNN, demonstrating that it can meet any specified target marginal coverage while drastically reducing prediction set or interval size by up to 74% when compared to baseline models. The overall presentation is easy-to-follow, and the technical contribution is non-trivial.

**Weaknesses:**

I don't see major flaws in this manuscript, but the introduction to how to construct correlation datasets can be more clear. In addition, the manuscript seems to miss a portion of works [1, 2, 3] on quantifying the uncertainty in GNN predictions. If the proposal of conformal prediction is to quantifying the uncertainty of GNN prediction, then other approaches on uncertainty quantification of node classification with GNNs should not be neglected.

[1] Stadler, Maximilian, et al. "Graph posterior network: Bayesian predictive uncertainty for node classification." Advances in Neural Information Processing Systems 34 (2021): 18033-18048.
[2] Zhao, Xujiang, et al. "Uncertainty aware semi-supervised learning on graph data." Advances in Neural Information Processing Systems 33 (2020): 12827-12836.
[3] Gao, Jiayi, et al. "Topology Uncertainty Modeling For Imbalanced Node Classification on Graphs." ICASSP 2023-2023 IEEE International Conference on Acoustics, Speech and Signal Processing (ICASSP). IEEE, 2023.

**Questions:**

Please refer to the Weakness section.

**Limitations:**

I am not seeing limitations.

---

> ### Author Rebuttal · Authors · 2023-08-09
>
> We thank the reviewer for insightful feedback and for noting that our paper is interesting, timely, and easy to follow. The reviewer raises great questions and we respond to them below:
>
> > Clarification on the construction of the correction dataset
>
> We thank the reviewer for raising this issue, which is helpful in improving the clarity of our manuscript. We shall clarify in the revision. The correction dataset was randomly sampled from the calibration dataset with a pre-specified sample size. It plays a crucial role in our approach, enabling the calculation of inefficiency loss by simulating the downstream conformal steps.
>
> > Inclusion of GNN-based uncertainty quantification methods.
>
> We thank the reviewer for pointing us to these papers. We will discuss them in the expanded section on related works. Note that the methods in question are not directly comparable to our work since they produce an uncertainty score representing the model uncertainty per node, which aims at OOD/misclassification detection. As a result, these works do not construct uncertainty scores for all classes and do not produce prediction intervals/sets. In contrast, we aim to construct a prediction set/interval and require a score for every class. We also want to mention that conformal prediction could also be extended to these problems in non-graph settings [1,2] and we happen to be interested in extending it to graph OOD detections. However, this direction is out of the scope of this current work.
>
> [1] Kaur, Ramneet, et al. "iDECODe: In-distribution equivariance for conformal out-of-distribution detection." Proceedings of the AAAI Conference on Artificial Intelligence. Vol. 36. No. 7. 2022.
>
> [2] Ishimtsev, Vladislav, et al. "Conformal $ k $-NN Anomaly Detector for Univariate Data Streams." Conformal and Probabilistic Prediction and Applications. PMLR, 2017.

---

### Official Review · Reviewer_M4K7 · 2023-07-07

**Soundness:** 4 excellent
**Presentation:** 4 excellent
**Contribution:** 4 excellent
**Rating:** 8
**Confidence:** 5

**Summary:**

This paper proposes a conformal prediction method tailored for graph-structured data. The proposed correction method is topology-aware and based on an empirical observation that inefficiencies correlate highly with network edges. The method updates node predictions based on its neighbors, and it is trainable alongside the GNN model. They also show how regular conformal prediction methods work under

**Strengths:**

[1] The well-motivated problem, clean writing, and detailed related works.

[2] The first conformal prediction method for graph-structured data with exchangeability and validity assumptions is GNN agnostic and intuitive.

[3] Rigorous proof and method to show exchangeability and validity of conformal prediction on graph-structured data for the first time in the literature.

[4] Capable of achieving conditional coverage, which is a stronger version of marginal coverage.

[5] In-depth experiments and ablation studies show the efficacy and efficiency of the proposed method.

**Weaknesses:**

[1] Inductive settings for GNN problems are more realistic compared to transductive settings. It is also not motivated why authors start with a transductive setting.

[2] There could be ablation studies/experiments over the conformity score functions, such as testing with RAPS[1].

[3] Even though the original coverage definition is used to measure inefficiency, for graph-structured data, there is an inherent non-IIDness in the data. Therefore each sample having equal weights in the coverage calculation is not appropriate. Weighting based on the degree of a node could be a great idea. Would love to discuss this part during the rebuttals.

[4] Why is there no experimental comparison with DAPS[2] (ICML23), which is also a conformal prediction method for node prediction?

[5] The paper's assumption of exchangeability is strict. What happens if exchangeability does not hold?


[1] Angelopoulos, Anastasios, et al. "Uncertainty sets for image classifiers using conformal prediction." arXiv preprint arXiv:2009.14193 (2020).

[2[ Zargarbashi, Soroush H., Simone Antonelli, and Aleksandar Bojchevski. "Conformal Prediction Sets for Graph Neural Networks." (2023).

**Questions:**

(1) Why do authors not tackle with inductive node classification problem?

(2) What happens if permutation invariance does not hold?

(3) What is the reason for not using RAPS[1]?

(5) In lines 133-134, while employing conformal prediction. is it even possible to change the trained prediction?

(6) My intuitive understanding is that if inefficiency is correlated with the network edges, why degree does not affect the coverage? I feel that coverage for graph data should be redefined for graph-structured data. For example, each sample could be weighted based on the degree of the node.

(7) In what circumstances does exchangeability not hold?


[1] Angelopoulos, Anastasios, et al. "Uncertainty sets for image classifiers using conformal prediction." arXiv preprint arXiv:2009.14193 (2020).

**Limitations:**

The authors have addressed limitations perfectly. As they also mentioned, this method is valid for transductive settings, but it will not be optimal for inductive settings. Authors also plan to extend it for inductive settings and link prediction tasks. Also, this method is heavily based on the exchangeability assumptions. It is unclear how to modify the algorithm because exchangeability assumptions do not hold.

---

> ### Author Rebuttal · Authors · 2023-08-09
>
> We thank the reviewer for insightful feedback and for noting that our paper tackles a well-motivated problem, that our method is rigorous and novel, and that our experiments are in-depth. Below, we address the excellent questions raised, with numbers corresponding to those in the review (e.g., [W1] refers to Weakness 1, [Q1] to Question 1, etc.) and we combine similar issues as needed.
>
> > [W1, Q1] Why focus on the transductive setting? How about the inductive setting?
>
> Thank you for this great comment. The transductive setting is a widely used evaluation setup for graph machine learning [1]. In Section 3, we show in theory that graph exchangeability holds for transductive settings, which enables the application of conformal prediction to graph-structured data. However, in the inductive setting, freshly arrived testing nodes will be potentially connected to calibration nodes in the graph. Thus the non-conformity scores of calibration nodes are dependent on the ordering of testing nodes. In other words, different orderings of testing nodes that connect to the graph yield calibration nodes with different non-conformity scores. This violates the permutation invariance condition we laid out for graph exchangeability. This setting would be far from common conformal prediction approaches. Thus, in order to establish the foundation for graph conformal prediction, we focus on the transductive setting in this work and leave the extension to the inductive setting for future work.
>
> [1] Kipf and Welling. Semi-supervised classification with graph convolutional networks. ICLR 2016.
>
> > [Q2] What happens when the permutation invariance condition does not hold?
>
> Thank you for raising this question. Without any further assumptions, violating the permutation invariance condition may violate exchangeability and make conformal prediction invalid. On the other hand, permutation invariance is a sufficient, but not necessary, condition for exchangeability (which is critical for the validity of conformal prediction). There might exist scenarios where permutation invariance is violated but exchangeability still holds; we believe that investigation on such cases is an exciting direction that requires considerable additional efforts.
>
> > [W5, Q7] When does exchangeability not hold and what will happen?
>
> Thank you for this question. Exchangeability is a fundamental assumption for conformal prediction. In summary, there are two scenarios in the graph setting where exchangeability does not hold. One is the transductive setting with a non-random split. The second is the inductive setting.
>
> If it is violated, then conformal prediction will generally be invalid. One direction is to extend to the “beyond exchangeability” framework [1] by reweighting conformity scores. But adapting this to non-exchangeable graph data remains a challenge since it does not allow for data-dependent weights (e.g. those based on network features). We believe that exploring graph conformal prediction beyond exchangeability is an important (and admittedly challenging) direction to pursue, which we leave for future research.
>
> [1] Barber et al. Conformal prediction beyond exchangeability. The Annals of Statistics 2023.
>
> > [W2, Q3] What is the result for non-conformity score RAPS?
>
> Thanks for this suggestion. CF-GNN is agnostic to the choice of non-conformity score. We have picked CQR and APS since they are representative choices for regression and classification. Following the suggestion, we further experimented on RAPS and showed that CF-GNN can still obtain consistent improvement in efficiency reduction:
>
> |Size|CP|CF-GNN|%Improvement|
> |-|-|-|-|
> |Cora|1.67±0.11|1.42±0.05|-15.35%|
> |DBLP|1.39±0.02|1.32±0.01|-5.00%|
> |CiteSeer|1.30±0.07|1.04±0.04|-19.85%|
>
> > [W4] Comparison with DAPS.
>
> Thank you for pointing out this interesting paper. We would like to first emphasize that the *DAPS is concurrent work, and that it was made public after we submitted our work to NeurIPS*. This is why we were not able to compare in our submission. We also note that DAPS employs diffusion to aggregate non-conformity scores, whereas CF-GNN introduces a learnable framework that optimizes efficiency through a specialized loss function. In fact, CF-GNN can be seen as a strict generalization of DAPS, becoming equivalent when employing a simple sum function over immediate neighbors. Our method's versatility also extends to regression tasks, highlighting further distinctions from DAPS. We’ll discuss it in the paper.
>
> > [W3, Q6] Redefining degree-weighted coverage?
>
> Thank you for raising this intriguing question. We first note that modifying the coverage definition with degrees would incur substantial modifications to the theory underlying graph conformal prediction. However, even without changing the definition, conditional coverage based on node degree can give insights into what we would get for your "weighted" coverage. The idea is that if the conditional coverage is good across all values of node degree, then degree-weighted coverage is also valid. As shown in Supplementary Table 9, with base CP, the worst-slice conditional coverage (WSC) conditioned on degree decreases to 0.79 when the target is 0.9, supporting the reviewer’s claim that marginal coverage may neglect variation in node degrees. In contrast, our CF-GNN has 0.89 WSC, implying satisfactory degree-based coverage. We hypothesize that it is due to network smoothing, where the prediction of low-degree nodes is modified by connections to high-degree nodes. More theoretical and empirical analysis will be interesting and left for future work.
>
> > [Q5] Tradeoff between uncertainty quantification and predictive performance?
>
> We thank the reviewer for bringing up this important question. Reviewer Wep3 has also expressed interest in this matter. Due to the constraints on the length of our response to individual reviewers, we direct the reviewer to our detailed answer provided in Reviewer Wep3's inquiries [Question 2].

---

### Official Review · Reviewer_Wep3 · 2023-07-27

**Soundness:** 3 good
**Presentation:** 3 good
**Contribution:** 3 good
**Rating:** 6
**Confidence:** 3

**Summary:**

This paper studies the problem of providing faithful and "efficient" uncertainty estimates for GNNs. Here faithful means the unknown groundtruth is contained in the prediction set with a probability higher than a threshold; efficient means the prediction set should be as small as possible. Specifically, the proposed method is based on the conformal prediction method, which uses a separate calibration set to determine the threshold used to decide whether a class (for classification) is included in the prediction set.

The authors first show that conformal prediction can be applied to GNNs as long as the samples in the calibration and test sets are exchangeable. This further leads to the observation that the test time coverage of the predictions fluctuates a lot when the number of test samples is small. The authors term this problem as the "inefficiency" during conformal prediction.

To improve both faithfulness and efficiency of the uncertainty estimates, the authors propose to train a new GNN that uses the original GNN's predictions as input and adjusted predictions as output. Instead of employing the original predictive loss, the paper proposes a loss that is a differentiable proxy of the efficiency metric. This model is trained with a pseudo calibration and test set split from the original validation set.

Experiments show better uncertainty estimates compared to well-known approaches such as MC dropout, although the baselines are not designed specifically for GNNs.

**Strengths:**

- The paper formally justifies that conformal prediction can be used under the exchangeability assumption.

- The proposed method is simple yet effective: directly optimizing for better uncertainty quantification on a potentially much smaller model.

**Weaknesses:**

- It seems that the calibration GNN needs to be trained for every $\alpha$ separately if we want to have multiple calibration thresholds. It would be nice to show whether it is possible to train a model that adapts to multiple $\alpha$s.

- It would be useful to show the tradeoff between the predictive performance and the uncertainty quantification performance. For example, the change in top-1 accuracy of the calibrated and original prediction.

- Could the calibration GNN suffer from overfitting? In an extreme case, if $\mathcal{V}_{\mathrm{cor \_cal}}$  and $\mathcal{V}_{\mathrm{cor \_test}}$ are both small, it might be possible that the calibration GNN minimizes its loss by overfitting the labels and make all Vs trivially 0? Maybe this is one reason for only giving the calibration GNN the output of the base GNN?


**Questions:**

- To evaluate $\hat{\eta}$, all samples from the calibration set are needed. If the graph is too large to be processed in a single batch, is it possible to train the calibration GNN in a mini-batch fashion? How would this affect the performance?

- It would be nice to show the training cost of the proposed method. I would expect training the calibration GNN could be much faster, but it is better to have more quantitative results.

**Limitations:**

I do not see potential negative societal impact of the work.

---

> ### Author Rebuttal · Authors · 2023-08-09
>
> We thank the reviewer for the constructive feedback, and for recognizing that our approach is simple yet effective. We appreciate the thoughtful questions posed and address them in detail below:
>
> > [1] Train a separate model for every $\alpha$?
>
> The reviewer recognizes a vital part of our approach, where CF-GNN does require a separate model for each $\alpha$ because of the predefined nature of $\alpha$ in conformal prediction. We acknowledge that an extension to accommodate multiple $\alpha$s simultaneously could augment our method's flexibility; this is an area for future research.
>
> In our calibration GNN, the chosen $\alpha$ plays a role in the loss function. A solution to include multiple $\alpha$s might be to aggregate the losses for these $\alpha$s. This will make the calibration GNN universal for multiple $\alpha$ values, but possibly suboptimal when compared to a model trained for a specific $\alpha$ value.
>
> It's crucial to note that in most real-world scenarios, the miscoverage level $\alpha$ is often fixed (e.g., 5%), so training multiple models isn't typically a major concern, as only one or a limited number of $\alpha$ values are usually explored. This makes our CF-GNN approach practical. This aspect will be thoroughly discussed in our revised manuscript.
>
> > [2] Tradeoff between uncertainty quantification and predictive performance?
>
> Thank you for posing this insightful question. From our understanding, uncertainty quantification refers to the prediction set and its size, whereas predictive performance relates to the precision of the point estimate. We will explain based on this understanding, but please feel free to correct us if you meant something else.
>
> For regression tasks using quantile regression, CF-GNN focuses on adjusting the $(\alpha/2, 1-\alpha/2)$ quantile band. Since the quantile bands are typically not used for point prediction, this adjustment does not influence the predictive performance of the model.
>
> In classification tasks, the correction step within CF-GNN could indeed modify the top-1 class prediction (we view this as the “point prediction”). However, this correction typically does not result in a visible change in accuracy. We show this with newly conducted experiments below:
>
> |Data|Before|After|
> |-|-|-|
> |Cora|0.844 ± 0.004|0.843 ± 0.016|
> |DBLP |0.835 ± 0.001|0.832 ± 0.002|
> |CiteSeer|0.913 ± 0.002|0.911 ± 0.002|
>
> The efficiency of prediction sets and the accuracy of point predictions are typically distinct goals requiring different optimal models. Our method employs a second calibration GNN to separate these goals, allowing the base GNN for point prediction and CF-GNN for uncertainty quantification. In a new experiment on Cora, we find that 100% of the top-1 class from the base GNN are in CF-GNN’s prediction sets. The potential to develop steps that explicitly consider point prediction accuracy is an exciting avenue for future research, and this will be discussed in the revised manuscript.
>
> > [3] Overfitting when the correction dataset is small? Possible to make all Vs trivially 0?
>
> Thank you for raising this pertinent question!
>
> **On the Issue of Trivially 0 Vs**: In classification tasks, the APS employs a cumulative sum of class probabilities over the softmax score up to the ground truth class; as the summation is one, it ensures that Vs will not be close to zero. In regression, a consistency loss is applied such that quantile bands are close to the original prediction thereby ensuring that Vs will not diminish to zero. Thus, our design safeguards against Vs becoming trivially 0.
>
> **On the Risk of Overfitting**: Indeed, when the size of the correction set is exceedingly small, the $\hat{\eta}$ estimate may be less accurate or too variable, which potentially leads to a bias in the adjustment of prediction scores. However, our empirical studies in Figure 5 (2) show that CF-GNN's efficiency remains robust across different calibration set sizes:
>
> |Holdout Calibration Set Fraction|CF-GNN|
> |--|--|
> |10% (36 nodes)|2.05±0.06|
> |30% (109 nodes)|2.14±0.12|
> |50% (182 nodes)|2.16±0.11|
>
> In addition, we emphasize that this does not impact the validity of conformal prediction. With a separate conformal prediction procedure that follows the correction step, we always guarantee valid coverage (this only relies on the exchangeability condition we derive), even in the face of potential overfitting in the correction step, and thus acts as an additional protection layer.
>
> > [4] What if the graph is large?
>
> Thank you for this insightful comment! We can modify the mini-batching procedure when the graph is large as follows: in each step, in addition to the sampled training set batch, we also make separate predictions for the correction calibration/testing set. These predictions are used to estimate $\hat{\eta}$ and calculate the inefficiency loss. Since we set the correction set to be min(1000, (|D_calib| + |D_test|)/2), when the graph is large, the correction set is set to be 1000, which incurs little computational overhead.
>
> Here we demonstrate its performance on the OGB-arXiv dataset which has 169,343 nodes and 1,166,243 edges. We observe it achieves consistent improvement over base CP on efficiency. We will discuss the large graph issue in the updated paper.
>
> | Method | Size |
> |-|-|
> |CP| 8.79±0.19 |
> |CF-GNN| 4.60±0.15|
>
> > [5] What is the training cost?
>
> Thanks for raising this. The calibration step scales similarly to standard GNN training, which is scalable with mini-batching and sampling. Also, CF-GNN has a small input node attribute size (# of classes for classification and 2 for regression), which is often much smaller than the node attributes in the original graph dataset (e.g. in OGB-arXiv, the node attribute size is 128). Using a single Nvidia 2080 Ti RTX 11GB and PyG, the time to train CF-GNN for the smallest graph Cora is ~3 minutes, and for the largest graph, OGB-arXiv is ~1 hour. We will discuss the scalability issue in the updated paper.

---

### Author Rebuttal · Authors · 2023-08-09

> Summary of main points

We thank the reviewers for their valuable feedback and constructive suggestions for improvement. Overall, all five reviewers considered our work well-written and well-motivated, and all appreciated the theoretical rigor and strong empirical performance of our proposed method.

A few stimulating questions raised have motivated us to conduct additional numerical experiments, which strengthen the empirical evidence and the paper. In summary, we found the following properties through new experiments:

- The training cost of calibration GNN is manageable as it scales well to large graphs using a mini-batching strategy (Wep3);
- Our framework consistently improves the efficiency for other nonconformity scores beyond APS and CQR (M4K7);
- Our framework has minimal impact on model accuracy (Wep3, M4K7);
- Our method is more efficient than other UQ methods when the latter is manually tuned to a specific coverage level (y5h3).

We also added further discussion in response to the reviewers’ important suggestions for clarification, including

- the motivation of the transductive setting (M4K7, rkWb)
- the importance of permutation invariance and exchangeability (M4K7)
- the coverage after weighting based on node degrees (M4K7)
- the robustness to overfitting (Wep3)
- the compatibility with mini-batch training (Wep3)
- the tradeoff between prediction accuracy and uncertainty quantification (Wep3, M4K7)
- relation with related works on GNN-based uncertainty quantification (M4K7, XvAj)
- efficiency of baseline UQ methods (y5h3)

Please see our point-by-point response to all questions raised by each reviewer’s comments below.

---

### Decision · Program_Chairs · 2023-09-21

**Decision:**

Accept (spotlight)

**Comment:**

The paper introduces an approach that utilizes conformal prediction under the exchangeability assumption for graph-structured data. The method optimizes uncertainty quantification, offering simplicity and effectiveness. The paper presents a well-motivated problem, clear writing, and extensive related work. Notably, it's the first GNN-agnostic conformal prediction approach for graphs with exchangeability and validity assurances, backed by rigorous proofs. The method achieves conditional coverage, surpassing marginal coverage, as demonstrated through thorough experiments and ablation studies. The reviewers are in agreement that the paper should be accepted. We ask the authors to incorporate the points raised during the discussion such as additional related work.